

# 1 Development of high-resolution multi-scale modelling system

# 2 for simulation of coastal-fluvial urban flooding

Agnieszka Indiana Olbert[1], Joanne Comer[1], Stephen Nash[1], Michael Hartnett[1]
[1]Civil Engineering, College of Engineering and Informatics, Ryan Institute, National University of Ireland,
Galway, University Road, Galway, Ireland
*Correspondence t*o: Dr. Agnieszka Indiana Olbert (indiana.olbert@nuigalway.ie)
**Abstract**. Urban developments in coastal zones are often exposed to natural hazards such as flooding. In this
research, a state-of-the-art, multi-scale nested flood (MSN_Flood) model is applied to simulate complex coastal-
fluvial urban flooding due to combined effects of tides, surges and river discharges. Cork City on Ireland's
southwest coast is a study case. The flood modelling system comprises of a cascade of four dynamically linked
models that resolve the hydrodynamics of Cork Harbour and/or its sub region at four scales 90m, 30m, 6m and
2m.
Results demonstrate that the internalisation of the nested boundary through a use of ghost cells combined with a
tailored adaptive interpolation technique creates a highly dynamic moving boundary that permits flooding and
drying of the nested boundary. This novel feature of MSN_Flood provides a high degree of choice regarding the
location of the boundaries to the nested domain and therefore flexibility in model application. The nested
MSN_Flood model through dynamic downscaling facilitates significant improvements in accuracy of model
output without incurring the computational expense of high spatial resolution over the entire model domain. The
urban flood model provides full characteristics of water levels and flow regimes necessary for flood hazard
identification and flood risk assessment.
Keywords: Urban flooding; Coastal flooding; Fluvial flooding; Hydrodynamic modelling; Nesting; Moving
boundary



## 1 Introduction

Low lying developments in coastal zones are exposed to natural hazards such as storm surges, waves, tsunamis and/or high river flows which can lead to significant flooding. Coastal flooding can result in substantial economic and social impacts including loss of life, damage to property and disruption of essential services (Brown et al. 2007).

Coastal flooding results from a rise of sea water level above normal predicted tide level. On the European Continental Shelf, coastal flooding is associated with storms generated in the Atlantic Ocean that travel through, or in proximity to, the shelf. Storm surges are important consequences of these storms – a temporary water setup resulting from synoptic variation of atmospheric pressure and strong winds blowing towards the shelf causing water to pile up against the coast. Surge physics is well understood in principle (Ponte, 1994); the mechanism of its propagation on the European continental shelf as a response to meteorological conditions (wind stress and atmospheric pressure signal separate) has been explained by Olbert and Hartnett (2010).

Flood dynamics due to a combination of multiple process drivers such as tides, surges and river inflows and their interactions is extremely difficult to understand using non-modelling methods (Robins et al, 2011). In recent years the amount of flood modelling work has risen dramatically. Yet the modelling still encounters various problems of which  input data such as topography (Mason et al, 2007; Smith, 2002), mesh resolution (Sanders et al, 2010; Fewtrell, 2011; Horritt et al, 2006; Yu and Lane, 2006), bottom roughness (Mason et al., 2003; Horritt, 2000)  or modelling framework (Hunter et al., 2008) are of greatest challenge. So far, one of the main issues hampering research into coastal flood modelling has been the lack of topographic data of sufficiently high resolution and accuracy along with highly resolvable efficient models. In the past decade,  high resolution topographic has become more available with airborne scanning laser altimetry (LIDAR) technology (Gomes-Pereira and Wicherson, 1999) providing high resolution digital surface maps that can be used as model bathymetry (Marks and Bates, 2000). Although there are still problems with mapping urban areas and considerable post-processing is necessary to extract digital terrain model from digital surface model (Mason at al., 2007), the hydraulic/hydrodynamic models developed using LIDAR data allow them to numerically propagate surge and tidal waves into coastal areas. Model accuracy and computational cost are still issues to be addressed.

The most common and simple approach to the modelling of coastal flooding in urban areas is to link (externally or dynamically) longitudinal 1D or latterly averaged 2D hydraulic models with coastal models (e.g. Formaggia, 2001; Chen, 2007; Brown et al., 2007). Such a set up has two significant drawbacks. Firstly, 1D/2D hydraulic



models work with the assumption that the lateral variations in velocity magnitudes are small, while in reality
many coastal floodplains (e.g. urban areas) contain channels that have a significant influence on the
development of inundation by providing routes along which storm surges propagate inland (Bates et al., 2005)
and therefore may lead to misrepresentation of localized flooding (Cook and Merwade, 2009; Mark et al, 2004).
Secondly, numerical errors may be introduced when linking different models with different dimensions resulting
from poor conservation of momentum (Yang et al., 2012). There is evidence of proven difficulty in ensuring
that each model interprets the model inputs and boundary conditions in the same way (Hunter et al. 2008;
Pender and Neelz, 2010).
These problems may be overcome by application of a single hydrodynamic model to both coastal waters and
coastal floodplains. Although such a model would allow smooth transition of the model solution between
coastal waters and floodplains, the full solution at scales appropriate for flood inundation would incur a
significant computational cost. On one hand, such models need to extend far enough offshore to capture the
development and propagation of surge and to resolve the nonlinear shallow water dynamics (interactions
between tides, surges and waves) at a resolution that is commensurate with flow features. On the other hand the
model needs to include upstream river channels, tidal flats, low-lying land and urban areas which are susceptible
to flooding at very fine resolution. This often results in a model setup that requires a large computational domain
of which the area of particular interest (such as floodplains here) comprises only a small percentage. For
structured grid models such requirements are often cost prohibitive and the alternative is to use lower resolution
at the expense of accuracy. This means that model discretization is performed at scales well below those
achievable with LIDAR data (the level of individual buildings in the case of urban flooding) meaning the
highly-resolved LIDAR data are not being optimally used (McMillian and Brasington, 2007). Some quite
successful attempts have been made using unstructured-grid models allowing selective grid refinement (e.g.
Yang et al. 2012; Robins et al., 2011); however, the computational demand of these models is high. A relatively
new approach to address this problem in high-resolution flood modelling makes use of continuing advances in
computational resources through numerical domain decomposition and multi core architecture runs (Sanders et
al., 2010). This method, however, requires substantial computational resources not commonly available yet.
In reality the modelling of coastal flooding (particularly in an urban environment) is a multi-scale problem that
requires accurate solution at various scales ranging from coastal sea or estuary scale down to a dense street
network of the inundated urban area. In the case of single rectilinear grid models, which are still the most
commonly used hydrodynamic models, this spatial resolution problem may be overcome by grid nesting; this





involves embedding higher resolution grids within a lower resolution global large-scale grid model. Such a
solution allows users to specify high resolution in a sub-region of the model domain without incurring the
computational expense of fine resolution over the entire domain. Nonetheless, the nested model for simulation
of floodplains must be very carefully chosen due to the flooding and drying properties of such zones; most
nested models developed to date do not incorporate flooding and drying as they have been developed
specifically for large-scale application where this phenomenon is not important (e.g. ROMS, Haidvogel et al.,
2008) or, even if they incorporate flooding and drying such as Mike21 (DHI Software, 2001) flooding and
drying of open boundaries is prohibited. This problem has been recently resolved in the multi-scale nested flood
(MSN_Flood) model of Nash and Hartnett (2010) which allows flooding and drying both within the domain and
along boundaries, while maintaining accuracy and computational efficiency. This model is ideally suited for
high-resolution modelling of urban flooding and, therefore, has been adopted for further development in this
research.
In this context, the authors present in this paper for the first time the application of the state-of-the-art flood
model, MSN_Flood, to complex coastal-fluvial urban flooding in the estuary-lying Cork City which is subject to
the combined effects of tides, surges and river discharges. The primary objectives of this paper then are to
present the development of this model and to critically examine its capability to forecast/hindcast the urban
inundation. It will be demonstrated in this paper that through the novel solution to the nested boundary, the so-
called moving boundary, the nested model allows simulation of the propagation of open sea conditions up to the
tidally active river upstream as well as rural and urban floodplains in a computationally efficient manner without
compromising model accuracy or stability.
The modelling framework proposed in this research comprises of a cascade of multiple nested models that
dynamically downscale large scale, coastal sea processes to the fine resolution scale of urban environments.
MSN_Flood was applied to the area of Cork City, Ireland, and its coastal floodplains; Cork City is frequently
subject to coastal-fluvial flooding. An extreme flood event of November 2009 that resulted in approximately
€100 million of flood damage in the city and its surrounds was chosen as a test case. The main features of this
accurate and efficient hydraulic modelling are illustrated through the Cork City application. In particular,
wetting and drying routine, computational efficiency and accuracy of simulated water elevations and velocity
fields are subject to in-depth analysis in this research.
This paper is organized as follows: section 1 describes the motivation for this research and related work; section
2 describes modelling, model setup and datasets; section 3 presents and compares numerical model results with



observed datasets; section 4 discusses the advantages  the MSN_Flood modelling system, and finally section 5
contains conclusions from the research.

**2  Methodology**
In this section a modelling system for coastal flood inundation is described along with the datasets and model
setup for the Cork City flood event.

**2.1 Modelling framework**
Many flood inundation events in urban environments have been modelled using simple hydraulic models (such
as HEC-RAS (Pappenberger et al. 2005) or LISFLOOD-FP (Bates and De Roo (2000)) incapable of simulating
flood water velocities required for accurate determination of flood wave propagation routes and assessment of
risks associated with a certain flood flow magnitude. A more realistic analysis can be achieved using a
hydrodynamic model that resolves both the continuity and momentum equations throughout the entire domain.
Here, the MSN_Flood model was applied to Cork City using a cascade of four nested grids to describe
hydrodynamics at various scales with particular interest in water elevations and velocity fields over the
inundated area. This nested model facilitates the refinement of spatial resolution in Cork Harbour from 90 m at
the outer reaches of the harbour down to 2 m in the streets of Cork City.

**2.2 Hydrodynamics**
MSN is a two-dimensional, depth-averaged, finite difference model and its solver is based on the alternate
direction implicit (ADI) solver developed by Falconer (1984) (Lin and Falconer, 1997, Nash and Hartnett,
2010). The governing differential equations used in the model to determine the water elevation and depth
integrated velocity fields in the horizontal plane are based on integrating the three-dimensional continuity and
Navier-Stokes equations over the water column depth. Assuming vertical accelerations are negligible compared
with gravity and that the Reynolds stresses in the vertical plane can be represented by a Boussinesq
approximation, then the depth integrated continuity and x-direction momentum equations are of the following
form (Falconer and Chen, 1991):





Continuity equation
$$\frac{\partial \zeta}{\partial t} + \frac{\partial q_x}{\partial x} + \frac{\partial q_y}{\partial y} = 0 \qquad (1)$$


Momentum equation in x-direction
$$\frac{\partial q_x}{\partial t} + \beta\left[\frac{\partial U q_x}{\partial x} + \frac{\partial U q_y}{\partial y}\right] = f q_y - gH\frac{\partial \zeta}{\partial x} + \frac{\rho_a C^* W_x (W_x^2 + W_y^2)^{1/2}}{\rho} \qquad (2)$$


$$-\frac{gU(U^2 + V^2)^{1/2}}{C^2} + 2\frac{\partial}{\partial x}\left[\varepsilon H\frac{\partial U}{\partial x}\right] + \frac{\partial}{\partial y}\left[\varepsilon H\left[\frac{\partial U}{\partial y} + \frac{\partial V}{\partial x}\right]\right]$$

where,   $t$        = time

$q_x, q_y$     = depth integrated volumetric flux components in the $x,y$ directions

$(q_x = UH, q_y = VH)$

$H$        = total water depth

$\beta$        = momentum correction factor for non-uniform vertical velocity profile

$f$        = Coriolis parameter (= $2\omega Sin\,\phi$, where $\omega$ = angular velocity of the earth's rotation and $\phi$ =

geographical latitude)

$g$        = gravitational acceleration

$\rho_a, \rho$     = air and fluid densities respectively

$C^*$        = air-water interfacial resistance coefficient

$W_x, W_y$     = wind velocity components in $x,y$ directions

C        = Chezy bed roughness coefficient

$\varepsilon$        = depth mean eddy viscosity



**2.3 Nesting structure and procedure**
MSN_Flood consists of one outer coarse grid called the parent grid (PG) into which one or more inner fine grids
(child grids, CG) are one-way nested. The model also enables multiple nesting such that a child grid may also be



a parent to another child. In this way, multi-scale nesting can be specified enabling high spatial resolution in
areas of interest. PG and CG models are dynamically coupled and synchronous. An overview of the nesting
procedure is schematically presented in Fig. 1. As can be seen, the time integration is a bottom-up approach
where PG can be advanced in time only when all of its children are integrated to the parent current time. The
ADI solution technique to solve the governing continuity and momentum equations requires the sub-division of
each timestep into two half-timesteps.  The nesting procedure, for each nesting level, is summarized in the
following 5 steps:
1.  integrate outermost parent grid one timestep ($t+\Delta t_p$)
2.  extract parent grid data and interpolate (spatially and temporally) along child grid boundary to next time

levels of child grid ($t+\frac{1}{2}\Delta t_c$)  and ($t+\Delta t_c$)

3.  integrate child grid one timestep ( $t+\Delta t_c$)
4.  repeat Steps 2 and 3 so that the child grid is synchronised to the current timestep of parent grid ($t+\Delta t_p$)
5.  return to Step 1 and continue.
The nesting procedure is similar in principle to other nested models (Holt et al., 2009; Korres and Lascaratos,
2003; Nittis et al., 2006) but the uniqueness of MSN_Flood is a novel approach to boundary formulation
through an incorporation of ghost cells in a manner that the nested boundary operates as an internal boundary.
Ghost cells (GC) are specified adjacent to nested boundaries so that the boundary configuration consist of two
rows/columns of CG cells: internal boundary cells and the adjacent exterior ghost cells. A schematic of the
general configuration of the nested boundary is shown in Fig. 2. In this internal boundary approach, PG
boundary data is specified to both the ghost cells outside the CG domain and to the internal boundary cells
allowing the governing equations of motion at the internal boundary grid cells to be formulated and solved in
the same way as interior grid cells. This enables accurate specification and conservation of incoming fluxes of
mass and momentum along the boundaries of the nests. To demonstrate befits of this approach the finite
difference formulation for the advective term in the momentum equation, which is key to momentum
conservation,  at boundary cells becomes:

$$\frac{\partial Uq_x}{\partial x} = \left[ \frac{[U(x+\Delta x, y)+U(x, y)]}{2} \cdot \frac{[q_x(x+\Delta x, y)+q_x(x, y)]}{2} \right.$$    (3)
$$\left. -\frac{[U(x, y)+U(x-\Delta x, y)]}{2} \cdot \frac{[q_x(x, y)+q_x(x-\Delta x, y)]}{2} \right]$$




For comparison, in a boundary formulation without ghost cells, the derivative $\partial U q_x / \partial x$ would be set to zero
as ghost cell grid points $U(x + \Delta x, y)$ or $U(x - \Delta x, y)$ would not exist, therefore momentum would not be
conserved between parent grid and child grid.
An important feature of the nesting approach in MSN_Flood is the implementation of moving boundaries along
the boundary of the nested domains. The flooding and drying routine originally developed in by Falconer and
Chen (1991) is implemented in MSN_Flood; this boundary formulation allows the model to be applied to areas
of inter-tidal zone or coastal flooding where there is typically a considerable degree of alternate flooding and
drying throughout the domain. The flooding and drying routine by Falconer and Chen has been extensively
tested in laboratory conditions and natural waterbodies  and shown to be stable and robust. However, when the
nested boundary was subject to flooding and drying, despite the overall improvement in mass and momentum
conservation along the nested boundary, significant errors were found to occur near the boundary in areas of
flooding and drying. This problem was overcome by implementation of an adaptive interpolation scheme which
uses linear interpolation or zeroth-order interpolation depending on the status (wet or dry) and the configuration
of parent grids along the boundary interface. More details of the method can be found in Nash (2010).  This
adaptive interpolation in combination with ghost cell and internal boundary formulation ensures the stable
flooding and drying of boundary cells.
The ghost cell formulation of the boundary was found to significantly reduce boundary formulation errors, one
of three error sources in nested models as classified by Nash and Hartnett (2010). Boundary formulation errors
arise from simplification of mathematical formulation of the governing equations of motion at open boundary
grid cells. Two other sources of errors at the boundary interface are boundary specification errors and boundary
operation errors. While the former errors arise from incorrect boundary data, and can be minimised by locating
nested boundary in areas of high PG accuracy, boundary operator errors result from the use of an inadequate
interpolation schemes and/or boundary condition for prescribing PG data to the CG boundary and are more
challenging to reduce. During the course of model developments various interpolation schemes were tested
including a zeroth order scheme, a linear scheme, a mass-conserving quadratic scheme and an inverse distance
weighted scheme. The linear interpolation was found to be most accurate in both time and space and therefore
was implemented in the model (Nash, 2010). With regards to the boundary conditions, three different types of
boundary conditions were tested, namely: Dirichlet condition, flow relaxation condition and radiation condition.
Extensive numerical testing showed that the most stable and accurate model solution could be achieved by



implementing the Dirichlet boundary condition. Accuracies of various interpolation and boundary condition
schemes were analysed and compared in Nash and Hartnett (2014).
Reduction in boundary errors due to the accurate development of boundary operators and more accurate
mathematical formulation of the nested boundary yielded significant improvements in conservation of mass and
momentum between parent and child grids. This in turn improved model stability at the nested boundary and
CG accuracy. These features make MSN_Flood highly applicable to modelling complex coastal flooding events
as in the current test case, where the nested boundary is located in the flooding and drying zone, and therefore
its length changes dynamically throughout the flooding event. This non-continuous moving boundary feature is
the subject of in-depth investigation in this research.


**2.4 Study area description and model setup**
Cork Harbour, in the southwest of Ireland, is a shallow (average depth 8.4 m) meso-tidal estuary with typical
spring tide ranges of 4.2m. Return levels of tides for 2- and 100-year return periods are 4.45 m and 4.52 m
above chart datum, respectively, while surge residual return levels for the same return periods are 0.56 m and
0.85 m, respectively (Olbert and Hartnett, 2013). The Cork Harbour domain is presented in Fig. 3. Cork City is a
densely populated urban area of approximately 120,000 people, located at the mouth of the River Lee which
drains into Cork Harbour. Tidal components of flooding in Cork City are due to combinations of high
astronomical tides and storm surges generated in the open ocean and propagating into the Harbour and
throughout the city streets. The River Lee corridor flows from west to east along the post-glacial valley into the
Lee proper, through Cork City, into Lough Mahon, Cork Harbour and south into Atlantic Ocean. In the city, the
River Lee bifurcates into the north and south channels around the Mardyke area and merges again at the eastern
edge of the city. The river flows for 2- and 100-year return periods are 208.6 and 307.7 $m^3$/s, respectively
(Halcrow 2008). Sea water intrusion up the river is bounded by a weir located 8km upstream from the river
mouth.
MSN_Flood was used in this research to develop a coastal-urban hydraulic model capable of simulating fluvial
and coastal flooding in the Cork City. The model grid needs to be setup to include not only river channel and
urban floodplains but also offshore waters necessary to resolve the non-linear hydrodynamics. The Cork
Harbour/City model is therefore configured as a four level cascade of dynamically linked nested grids that
resolve the hydrodynamics of the region at spatial scales of 90m, 30m, 6m and 2m. Each coarser grid provides





boundary conditions to the next finer grid, i.e. the 90m grid provides boundary conditions to the30m grid and
the 30 m grid provides boundary conditions to the 6m grid, etc. Fig. 4a illustrates the extent of each grid and the
nesting structure, while Fig. 4b shows details of the high resolution 6m grid and the 2m urban flood grid.
The parent grid (PG90) representing the full domain of Cork Harbour was resolved at a grid spacing of 90m. At
3:1 nesting ratio, the first child grid (CG30), completely embedded within the parent model domain, has a grid
spacing of 30m. The CG30 model provides boundary conditions to a 6m grid (CG06) at a 5:1 nesting ratio. The
domains of CG30 and CG06 models only partially overlap. Water elevations computed on CG30 are passed to
the eastern boundary of CG06 while River Lee flow data are specified at the western boundary of CG06.
Finally, the ultra-high resolution 2m child grid (CG02) is entirely embedded within CG06 and is used to
simulate urban flooding of Cork City. The nesting ratios of 3:1 and 5:1 used in this setup are in line with nesting
ratios used in other studies (e.g. Spall and Holland, 1991). Configurations of the nested models are summarized
in Table 1.
Open boundary conditions to the MSN_Flood parent grid, PG90, are provided as total water elevations
containing tidal and surge signals extracted from an ocean model of the North East Atlantic (Olbert and
Hartnett, 2010). The surface boundary of the MSN_Flood model is forced by 10-m wind fields and mean sea
level atmospheric pressure obtained from the regional analysis ERA-40 model (Uppala et al., 2005) and
operational model first-guess dataset (Simmons et al., 1989). River Lee discharges from gauge station 19011
were provided by Office of Public Works (OPW), Ireland. Admiralty Chart data were used to develop the
bathymetric model of Cork Harbour, while high resolution LiDAR data provided by the OPW were used to
construct the high resolution urban digital bathymetric model. The channel of the River Lee was included in the
model based on cross-sectional survey data also provided by the OPW from an extensive survey of the River
Lee catchment in 2008.

**3 Results**
Showcasing the capability of the multilevel nesting integrated system to accurately simulate the extent and level
of urban flooding is central to this research. MSN_Flood has been extensively tested in both laboratory settings
(against physical tidal models) and natural open harbours. In this research, a comprehensive validation of the
model in a coastal flood application to Cork Harbour and the urban environment of Cork City is presented.
Initial evaluation of model accuracy is carried out at each of the four levels of nesting; both modelled water
elevations and velocities are compared to available field data. The assessment of the model skill in simulation of





urban flooding is carried out for the November 2009 coastal-fluvial flooding of Cork City. In this application,
the city streets and open areas are treated as hydraulics channels and plains that can be inundated depending on
the tide, surge and fluvial conditions. This is a highly complex hydrodynamic region to model and, therefore,
represents a robust test of the model.


**3.1 Validation of the nesting procedure**
**3.1.1 PG90 model**
Firstly, the performance of the low resolution 90m parent grid (PG90) model was assessed. Figure 5 compares
current velocities simulated by the PG90 model with measured data at Passage West in Cork Harbour over a
spring tidal cycle (see Fig. 3 for point P1 location). Results show that although pattern of currents through flood
and ebb conditions are relatively well predicted, the slack water conditions, where velocities are generally
smaller, are not reproduced correctly by the PG90 model. A higher resolution single grid (SG30) model at 30m
grid spacing was developed to test the accuracy of PG90. The same domain extents (Fig. 4) and the same
physical conditions were specified to the SG30 and PG90 models. As shown in Fig. 5 an increased resolution of
the model significantly improves model predictions throughout the tidal cycle and particularly during periods of
slack water.
The spatial distribution of PG model error was quantified by calculating the tidally-averaged relative errors
$RE_T$ which expresses a percentage error in a PG solution, $Y$ , relative to a higher resolution SG reference
solution, $X$ , at the output time $n$ over a tidal cycle (N=25)
$$RE_T = \frac{\sum_{n=1}^{N}|Y_n - X_n|}{\sum_{n=1}^{N}|X_n|} \cdot 100 \qquad (4)$$
Figure 6 shows the distribution of $RE_T$ in PG velocities in Cork Harbour; it can be seen that the errors generated
by the PG model are well over 30% at certain locations within the harbour (harbour entrance, along the
coastline, narrow channels and estuaries) so increasing the resolution from 90m to 30m leads to significant
reduction in the error. However, improvements in accuracy due to higher spatial resolution come at a high
computational cost which for the SG model (80min for 50hrs run) is nine times that of the PG model (9min for
50 hrs run). The use of nested model is then a justifiable and favourable solution.



In the course of extensive validation, the timeseries of PG90 and SG30 were also inter-compared. Figure 7
shows water elevations and current velocities in Lough Mahon (see Fig. 3 for point C1 location). Water
elevations computed by both models are in very good agreement. In contrast, current velocities are significantly
overpredicted by the PG90 model. Linear regression of current speeds of PG90 against SG30 solution is shown
in Fig. 8. As can be seen from this figure the correlation coefficient between PG90 and SG30 is 0.89 while slope
and intercept are m=1.24 and c=0.03, respectively.

**3.1.2 CG30 model**
The selection of a child grid domain configuration is sensitive to the location of boundaries that may affect the
overall stability and performance of the nested model solution. Suitable CG boundaries must be located in areas
of low PG inaccuracy and at a sufficient distance from the area of interest as location of the boundary close to
the area of interest may result in boundary errors propagating into the area causing the accuracy of the solution
to deteriorate. On the other hand, boundaries need to be sufficiently close to the area of interest in order to
minimize the domain size (computational cost).
The first level child grid, CG30, was located in the north-west part of Cork Harbour with the centrally located
Lough Mahon (directly feeding to the River Lee estuary) being the area of interest. The boundaries for the CG30
domain were chosen based on the $RE_T$ distribution plot for the PG90 current velocities presented in Fig. 6. The
upper section of Passage West, connecting Lough Mahon with Lower Harbour, was selected as a suitable
southern boundary (SB) due to its relatively low $RE_T$ while the closest suitable location for the eastern
boundary (EB) was at a much greater distance from Lough Mahon due to generally high PG inaccuracies in the
North Channel.
The accuracy of the CG30 boundary location  was assessed by comparing the net fluxes of mass and momentum
across the corresponding interfaces in the PG90, SG30 and CG30 models. Net fluxes were calculated normal to
boundaries. Mass and momentum fluxes through the SB and EB boundaries are compared in Fig. 9 and 10,
respectively. It can be seen that the predominant forcing-boundary for the CG30 domain is the SB boundary.
The tidally-averaged errors in PG90 fluxes relative to the SG30 were approximately 4% for both mass and
momentum indicating a high level of PG90 accuracy. At the EB boundary, the PG90 accuracy was slightly
lower resulting in error in PG90 mass flux of 5% and momentum flux of 10%. However, this boundary
accounted for a smaller portion of the total boundary forcing, and its distant location from the area of interest
allowed boundary errors more time to dissipate. The tidally-averaged errors in CG30 fluxes (both mass and





momentum) relative to PG90 fluxes were less that 2% at both boundaries, demonstrating high levels of
conservation from parent grid to child grid.
Relative error analysis was also carried out for the entire CG30 model domain with respect to water elevations
and velocities, and results of these analyses are summarized in Table 2. The domain-averaged relative error
$RE_D$ ($= RE_T / N$) in the PG90 water elevations relative to the SG30 were 5.9% while in the CG30 model
this error was reduced to 1.1%. The extent of the domains with $RE_T$ greater than 1% was 94% for PG90 and
28% for CG30. The absolute error defined as:
$$AE_T = \frac{\sum_{n=1}^{N} |Y_n - X_n|}{N}$$
(5)

was also calculated. $AE_T$ in water level significantly decreased from 8cm in the PG90 to 1.2 cm in the CG30.
In relation to current velocities, the $RE_D$ was reduced from a large value of 22.4% in PG90 to just 0.5% in
CG30; while $RE_T$ values exceeding 5% were found in 72% and 4% of the PG90 and CG30 domains,
respectively.
As shown in Fig. 7, timeseries of water elevations and current speed show very good agreement between SG30
and CG30 throughout the tidal cycle. This indicates significant improvement in the accuracy of velocity
computation using the high resolution nested CG30 and is verified by the linear regression analysis shown in
Fig. 8. The superiority of CG30 over PG90 model when compared to SG30 is clear and confirmed by a
correlation coefficient of 0.99 compared to 0.89. The slope and intercept were also improved for CG30 when
compared to PG90; with m=1.01 and c=-0.01 the CG30 against SG30 model solutions lie approximately on the
45° line.
These results demonstrate that the application of the nested high resolution model results in significant
improvement in the accuracy of the model solution over the lower resolution PG solution. Similar to the
improvement in model accuracy, an equally significant reduction in computational effort was achieved. For
example, the application of MSN_Flood model to level 1 domain nesting yields 21 minutes simulation time for
the PG90-CG30 model; this is contrasted by 80 minutes simulation time for the SG30 model. Thus the nested
model runs 3.8 times quicker than the single grid model.




### 3.1.3 CG06 model

In contrast to the CG30 grid being fully embedded within the PG90 grid, in the second level of nesting CG06 is

only partially nested within its parent CG30 (Fig. 4). Approximately 38% of wet cells in CG06 overlap CG30.

This is a hybrid boundary structure where the east boundary is prescribed using hydrodynamic data from the

parent model while the west boundary is prescribed using measured data. The west boundary is a flow

boundary, with River Lee inflows extracted from river gauging station 19011. The east boundary is a water

elevation boundary where water elevations are supplied along the boundary by the CG30 model. The location of

the latter boundary was selected to correspond to the position of the Tivoli tidal gauge station and therefore to

contribute to model validation (see Fig. 3 for location of Tivoli gauge).

Validation of the CG06 model is conducted for the flood event of November 2009, which due to a combination

of heavy river discharges and high tides coinciding with moderate surges resulted in extensive inundation of the

area delineated by this nested grid. Figure 11 compares timeseries of water elevation computed at the CG30-

CG06 nested boundary (east boundary) against tidal gauge records from the same location. Overall, there is a

very good agreement between predicted water elevations and measured data. The high degree of model accuracy

is manifested by high correlation (0.992) and a low value of RMS difference (0.022m) shown in Table 3 (model

CG06_1). Both the RMSE (0.142m) and centred RMSD (0.141m) indicate that the model is able to reproduce

variability of water elevation with a good accuracy (order 0.14m). Further, a small difference between these two

statistical measures implies that the mean values of observations and simulation are very close. Interestingly, the

accuracy of the CG06 model is improved when a 6 minutes phase shift (one record timestep) between

observations and simulation is artificially introduced (model CG06_2 in Table 3). This results in RMSE

(RMSD) reduction to 0.106m (0.104m) and an increase of correlation to 0.996. It is deemed then that there is a

phase lag between model and observations of approximately one observational timestep. Another aspect of the

analysis involved temporal occurrence of an error. As the model-observations discrepancies are observed around

low water levels (which is not so significant to this study), by not considering negative water elevations (below

0 mOD Malin) the RMSE is further reduced to 0.075m (model CG06_3 in Table 3). Such level of agreement

between model and observation is considered to be satisfactory.

The effect of horizontal resolution on model skill is also examined. This is carried out by comparing the model

performance at 6 m and 2m resolutions.  For this purpose a single grid 2m reference model (SG02) covering the

area delineated by the CG06 model was developed.  Figure 12 presents the distribution of water level $RE_T$ in

the CG06 solution relative to the SG02 reference solution. In general, errors in CG06 outside the Cork City



centre are very low (<10%) implying that flooding in the rural area of Cork is well resolved using the 6m grid.
In contrast, significantly higher errors are obtained in the Cork City (CG02 domain), and in particularly in areas
of narrow dense streets where errors exceed 30%. Here, an increase in model resolution leads to a significant
reduction in errors. This implies that next level of nesting is required to improve the model accuracy in the city
centre.

**3.1.4 CG02 model**
Finally, the highest resolution 2m model (CG02), fully embedded within CG06, covers the urban area of Cork
City; this area is particularly prone to flooding. In the first step of model skill analysis, water elevations
simulated by the CG06 and CG02 models at four locations along the river channel are compared in Fig. 13 and
statistically summarized in Table 4. Again, the November 2009 flood event was used as a benchmark. Close to
the east boundary, at point CG02_4 (see Fig. 14 for point location), both models perform almost identical and
this is visually and statistically confirmed in Fig. 13d and in Table 4, respectively. Discrepancies between the
CG06 and CG02 models increase with distance from the nested east boundary and are manifested by overall
higher water elevations computed by the coarser CG06 model. Location CG02_2 (Fig. 13b) shows the biggest
discrepancy evidenced by the statistical measures RMSE=0.195m, RMSD=0.109m, RMSdiff=-0.181m. Despite
overprediction of water elevations by the CG06 model, the general water level trends in the two models are  in
good  agreement  (COR=0.997). Another  important  advantage  of  a  high  resolution  model  is  an  improved
numerical stability of the model solution. As can be seen from Fig. 13 a-c, some infrequent random oscillations
in water levels occurring in CG06 from numerical instability due to insufficient grid resolution are not present in
the finer CG02 model.
The effect of improved horizontal resolution is analysed spatially by means of $RE_T$ distribution plots. As
shown in Fig. 12, the 2m resolution is essential to resolve small scale processes of complex urban area. Figure
14 compares $RE_T$ between CG02 and SG02. In general, $RE_T$ is quite low at 10% in the western part of the
city along river banks increasing in eastward direction to 20% in narrow streets of city centre. This is a
considerable  improvement  when compared to $RE_T$ in CG06 relative to SG02. Moreover, as CG02 achieves a
similar level of accuracy to SG02 the computational cost is significantly reduced and constitutes enormous 96%
saving.





From this analysis it can be seen that the CG06-CG02 nesting results in a model performance generally
comparable to the single grid SG02 model but at a significantly reduced computational cost when compared to
the single grid model.
The ultimate conclusion from the model validation is that MSN_Flood facilitates significant improvements in
model accuracy without incurring the computational expense of high spatial resolution over the entire model
domain. The model setup constitutes a rigorous test of model performance and on that basis it can be further
concluded that the model is applicable to situations where nested boundaries are located in complex urban
floodplains that periodically wet and dry.

**3.2 Urban flood modelling**
For most of the time, city streets are dry and rivers draining the hinterland are contained within well-defined
river banks or walls.  However, when extreme flood events occur rivers may burst their banks and the city
streets become water conveyance channels. The simulation of the hydrodynamics associated with rapid urban
flood events is complex; many significant issues must be addressed such as flooding and drying, spatial
resolution, domain definition, frictional resistance and boundary descriptions. When modelling flood events, the
mathematical formulation of the nested boundaries that permit flooding and drying is of particular importance.
Also, the horizontal resolution necessary to resolve small scale processes must be considered. In particular,
these aspects of the MSN_Flood model will be discussed in this section.

**3.2.1 Extreme flood event**
On the 19[th] and 20[th] of November 2009 high River Lee flows combined with high astronomical tides and
moderate surge caused localized overtopping/breaching of the river banks resulting in widespread flooding of
Cork City. Evolution of the flood wave propagation simulated by the CG02 model is shown in Fig. 15.
Maximum flooding was reached at 9:30 on 20/11/2009 around the time of high tide and approximately 5 hours
after peak discharge of River Lee. At this juncture over 62ha of Cork City had been flooded. The most affected
zone was the city centre located between the north and south channels of the river; this area is a low-lying island
that over centuries was gradually reclaimed from marshland and it's low-lying topography combined with the
influence of river, estuary and harbour makes the area particularly vulnerable to flooding.
The accuracy of the urban inundation simulation was assessed against field observations of inundation extent
and maximum heights of flood waters. The observed and modelled ultimate extents of flooding in the city are


shown in the Fig. 16; the hindcasted extent of inundation matches very well that observed during the flood
event. With regards to flood level heights, observed water level marks were collected and post-processed by
OPW at 38 survey points across the flooded area; their distribution is shown in Fig. 17. The survey point data
were subsequently used to calibrate the model. Initial calibration tests showed that the model was most sensitive
to bottom roughness coefficient. An extensive statistical analysis of bed roughness parameterization was used to
provide an accurate model solution for flood inundation; details of that analysis are presented elsewhere. The
best fitting results (R=0.97, RMSD=0.26) were obtained for the following roughness values: upper
channel=0.90, lower channel=0.90, roads=0.1, city floodplain=0.1 and upstream floodplain=0.30. Figure 18
provides visual assessment of the best fit model skill; good agreement between the model and observations is
achieved as the model solution falls on the 45° line. Interestingly, better agreement was found for survey
locations in floodplains as opposed to points adjacent to the river bank. This could be attributed to the fact that
the majority of survey points are located away from the channel edge (many are actually at the floodplain edge).

### 3.3 Moving boundary

The specification of a nested boundary in a flood-prone area is particularly problematic; nested models
developed so far prohibit flooding and drying along open boundaries. This problem has been overcome in
MSN_Flood; its unique mathematical formulation of the nested boundary involving ghost cells, internal
boundary formulation and adaptive interpolation, ensures stable flooding and drying of boundary cells. In
MSN_Flood, any nested boundary can be placed within a flooding and drying zone and therefore may be subject
to significant lateral expansion and contraction. Moreover, the internalization of the boundary allows the
flooding and drying mechanism to approach the boundary of the nested domain from either upstream or
downstream. As the boundary alternatively floods or dries, the number of active boundary cells expands and
contracts accordingly. Depending on local topography, not only the length of the boundary may change but also
the number of active boundaries changes. Such a boundary is therefore a complex, non-continuous, moving
boundary that spatially and temporally changes its characteristics. This is a significant aspect of this research.
In the model setup, the urban CG02 model is entirely embedded within the CG06 model; mass and momentum
from the 6m model is transferred to the 2 m model via two nested boundaries – the western boundary
transferring River Lee waters from the upper to the lower channel of the river (it also geographically divides the
floodplains into upper and lower floodplains), and the eastern boundary exchanging waters with the estuary. The
western boundary of CG02 is located on the upstream fluvial floodplain which is prone to wetting and drying. A





cross section through this boundary illustrating the steep gradients of the river channel bathymetry and the
topography of the adjacent urban floodplains (which includes buildings) is shown in Fig. 19. The temporal
progression of water levels throughout the November 2009 flooding is also plotted. The reference water level at
simulation time t=4hr corresponds to a 187 m$^3$/s river flow (19$^{th}$ of November 2009 at 01:30). At this juncture
the flow greatly exceeds the average river flow of 40 m$^3$/s as it results from increased discharges from Inniscarra
dam. The storage capacity of Inniscarra Reservoir had been reached after a month-long period of record high
rainfalls and heavy downpours on the 18$^{th}$ and 19$^{th}$ of November. Over the course of the subsequent 28 hours
the discharges further increased to reach a maximum value of 560 m$^3$/s at 2:30 on November 20$^{th}$. The water
level at the boundary increased from 4.57 mOD at 22:30 November 18$^{th}$ to a peak of 5.74 mOD 28 hours later.
The extensive inundation of the upper channel floodplains (upstream floodplains) has a major effect on the
western boundary of the CG02 model. It can be seen in Fig. 19 that as the flooding progresses to a simulation
time of 8hrs a second wetted boundary is created south of the main channel boundary due to bifurcation of
flood waters into two channels (called here the main and side channels) approximately 1.2km upstream of the
boundary. Importantly, there is a significant difference in water elevation of 0.41m between the two channels of
the boundary. This results from the topography of the upstream floodplains and therefore local flow conditions.
The reason for the difference in water elevations along the two sections of the boundary can be explained with
the help of Fig. 20 showing three cross-sections including one (cross-section 3) located close to the nested
western boundary. As simulated by the CG06 model, downstream from cross-section 1, representing the
maximum cross-sectional extent of the inundated area, flood waters must flow around an elevated strip of rural
land and so splits at this point into two floodplain channels. This is shown in cross-section 2, located at mid
length of this 1 km long strip of land; here the water elevation difference between two channels is 0.31m. This
elevation difference further increases to 0.41m near the nested boundary (cross-section 3).
The temporal rise of water levels at a number of points across the western nested boundary is shown in Fig. 21.
Series A represents the main river channel, series B and C correspond to points adjacent to the river channel
while series D is located in the side channel. The difference in water elevations between the two boundaries is
apparent throughout the entire flooding period, though it is reduced with the progress of flooding.
An interesting characteristics of the moving boundary is it change in its length. As flood waters continue
overtopping the river banks, the area of inundation increases and is reflected in the elongation of the boundary.
The length of the main channel boundary is initially equal to the river width, this nearly doubles during flooding



as shown for t=12 hrs in Fig. 19. The temporal evolution of flooding through the boundary clearly demonstrates
that the nested boundary is a discontinuous moving boundary with a variable head.
The numerical stability of such dynamically changing properties of nested boundary is an important aspect of
nesting procedures.  Overall, a change in length as well as division into separate subsections does not markedly
impact computational stability nor model performance. In fact, as shown in Fig. 14, $RE_T$ computed over the
flooding period remain low within the CG02 domain despite significant changes to nested boundary
configurations and flow conditions.
As demonstrated in this section MSN_Flood is developed in a general-purpose manner that through stable and
accurate moving boundary provides a high degree of choice and flexibility regarding the location of the
boundaries to the nested domain.

**3.4 Model resolution**
Due to the highly irregular topography of urban environments and the highly dynamic flows involved, urban
flooding is a complex problem. Most of the flood models developed so far have focused on rural or semi
developed floodplains where isolated large structures can be modelled while small objects are ignored or
parameterized as bottom friction (Brown et al., 2007). Such modelling does not implicitly account for locking
effects of building on flow. As the presence of buildings may substantially increase flood extent when compared
with undeveloped floodplains the role of high resolution discretization is paramount. However, as Brown et al.,
(2007) found, the greatest source of modelling error with respect to grid resolution is associated with the
steepest gradients in topography which are susceptible to interpolation error.
Modelling of flood flow through urban area is difficult because of its need for stable and accurate solution of the
flow equation (Brown et al., 2007). Since accurate modelling requires a resolution commensurate with flow
features, dense street network flows through urban floodplains can only be fully resolved with a sufficiently
high resolution. However, satisfactory model resolution, and thus accuracy, incurs computational expense; a
balance between these two contradicting factors provides an optimal solution. Gallegos et al. (2009) found that a
5m resolution mesh that spans a street by approximately three cells achieves such balance. The characteristics of
urban residential areas of southern Californian  investigated in their study is different than that of an old
European development type towns comprising of narrow dense streets as Cork City. It follows that the 5m
model resolution is insufficient to resolve flow dynamics in such city centre street networks.





In order to analyse the overall effect of model resolution on simulation results, CG06 and CG02 model results
are compared. Visual comparison of flood inundation can be made from Fig. 22 which shows CG06 and CG02
model outputs representing the maximum extent of inundation during the November 2009 flooding. There is a
discrepancy in the extent and magnitude of flooding between the two models. Some zones and streets do not get
flooded in the CG06 model, which may be caused by the coarse representation of the street network and
associated lack of connectivity between certain streets, while in other zones flood water is present in areas
which remain dry according to observations and CG02 output. Figure 23 (a) shows the difference in water
elevations between CG02 and CG06 interpolated onto the 2m grid. It is clear that both the height and area of
flooding are affected. The absolute difference in water level is on average 0.13m and is underestimated by the
6m model by up to 0.4m  in the upper section of river and overestimated by approximately 0.3m in the lower
section. Figure 23 (b) shows a spatial distribution of RMSE between two models. There is a noticeable reduction
in model performance at coarser resolution of 0.08m RMSE over the entire domain and the error is generally
larger in the dense street network of the urbanized zone. Based on model results it is clear that a substantial
portion of the error  results from the coarse representation of topography since its gradient is greater that the
slope in water surface; however, some small portion of the error could be attributed to errors in LIDAR data
(~0.1m RMSE according to Bates et al, 2010) as well as interpolation from 6m down to 2m grid.
Another comparative measure involves a computation of relative differences in inundated area and flood water
volume between coarse and fine grid models expressed as a following ratio
$$RD = \frac{\left| X_f - X_c \right|}{X_f} \qquad (6)$$
Where $X$ is total inundated area or volume in the domain at a particular time whereas indices $c$ and $f$  denote
CG02 and CG06 solutions, respectively.
Figures 24 (a) and (b) show the evolution of differences in inundated areas and volumes throughout the
simulation. The significantly high relative difference in the area at the initial stage of flooding reaching 36% is
misleading as the relatively small total inundated area with a small flood time lag results in large discrepancies
at this stage (ca. 11ha). Nevertheless, when the flooding is more pronounced (over 30 ha, max 62.6ha) the
relative difference is still up to 10%. With regards to flood water volume in inundated areas the difference is
over 20% during first hours of flooding and still remains as high as 10% throughout the flood peak only  falling
to below 10% when the flood recedes. The total RMSE of inundated area and volume between 2m and 6m



models are 3.4ha and 21,367m$^3$. This comparison demonstrates that horizontal resolution is of paramount
importance when simulating flows through complex topography. It seems that for Cork City centre comprising
of dense network of narrow streets, neither the 5m resolution  requirement nor 3 cell street span would resolve
complex flood flow at satisfactory level of accuracy.

**3.5 Flood water velocities**
Another significant advantage of MSN_Flood is its ability to simulate the velocities of flood waters. As oppose
to simplified 2D hydraulic models frequently used in urban flooding, the hydrodynamic MSN_Flood includes
both the continuity and momentum equations, solving for both water elevations and water velocities. Figure 25
shows an example of flood water velocities computed by MSN_Flood in a selected area of Cork city centre
blown up for ease of viewing; one can see flood waters in both the river channel and the urban floodplain. This
zone is characterized by fast flowing shallow water subject to rapid transitions as it flows  down through the
steep section of recreational grounds adjacent to the river channel. The city downtown, in contrast, is a ponding
area with relatively stagnant waters.
Knowledge of velocity fields facilitates better understanding of flood water hydrodynamics and in particular the
mechanisms of flood propagation. The routes and speeds of flood waves provide important information for the
evaluation of flood risks to people's safety and to property, as well as to the planning and actions of emergency
response teams.

**4 Discussion**
Inundation of coastal areas due to coastal and/or fluvial urban flooding mechanisms is a very complex
hydrological phenomena, and developing a modelling system to accurately simulate it is not a trivial task. The
research presented in this paper demonstrates that the concept of nesting models is very suitable for complex
urban coastal flooding as they facilitate the development of an integrated system capable of resolving
hydrodynamics at spatial scales commensurate with flows and physical features of the region of interest. The
modelling system adopted here determines physical processes simultaneously at different scales ranging from
bay-size circulation (90 m) through mesoscale processes of coastal waters at 30 m resolution down to the ultra-
high scale environment of 2m. Validation results show that the model performs well at each of these scales.
The MSN_Flood model developed for use in this research is well suited for high resolution urban flood
simulation for a number of reasons. Firstly, it allows smooth transition of the model solution between coastal



waters and river floodplains while giving a very high level of conservation of mass and momentum between
parent and child grid (Nash and Hartnett, 2010). Through incorporation of ghost cells and formulation of a
dynamic internal boundary, MSN_Flood is designed to minimize boundary formulation error and therefore to
transfer mass and momentum across the nested boundary without loss of nested solution accuracy. The
reduction in boundary errors yields also a significant improvement in model stability at the nested boundary and
CG accuracy. This in turn permits stable flooding and drying at the boundary; moreover, these process are
allowed to approach the boundary of the nested domain from either upstream or downstream. The so-called
moving boundary allows then embedding of a child grid model within the parent model in areas where the
nested boundary may wet or dry making the model highly flexible in application. Interestingly, such highly
reduced boundary formulation errors is achieved  in a nesting mechanism where the nested boundary comprises
of only two cells of columns or rows (ghost cells and internal boundary cells). For comparison, in many nested
models  poor accuracy due to boundary formulation errors is commonly compensated  by indirect solutions such
as boundary configuration (e.g. location). For example, Kashefipour et al. (2002) in order to reduce possible
nesting error dynamically link  2D coastal model with 1D river model by using overlapping grids at the
boundary – a common area where  boundary values are exchanged between two models. Such model setup is
not required in MSN_Flood where accurate exchange of boundary conditions occurs along a boundary.
Secondly, the model has virtually no limit to the number of specified nesting levels (and spatial resolution) and
is primarily constrained by computational effort rather than numerical stability. The highest resolution of 2 m set
for this study was dictated solely by the resolution of available LiDAR data and higher resolutions are easily
achievable if suitable terrain data is available. For example, a 0.025 m resolution was used to simulate flows
corresponding to those in a physical scale model of a harbour of dimensions 1.0x1.0x0.25 m (Nash and Hartnett
2014). In this way, the model allows improved accuracy of solution when compared to a lower resolution parent
model where the improved accuracy is similar to that of a similar high resolution single grid model but the
computational effort is significantly reduced.
Thirdly, the model provides adequate solutions at scales sufficient for processes of interest, such as coarse
resolution coastal circulation and fine resolution flood inundation. This is attributed to the robust hydrodynamic
module which in essence adopts the well-tested numerical scheme and discretisation methods described by
Falconer and Chen (1991). The uniqueness and improvement of MSN_Flood over other nested models is its
formulation of the nested boundary in the area where flooding and drying may occur. In order to accommodate
flooding and drying of boundary cells the model allows a moving nested boundary so that large sections of the



boundary can alternatively wet and dry. The stable flooding and drying of boundary cells results from the
internalisation of the nested boundary combined with an adaptive interpolation technique tailored specifically
for this model. To the author's knowledge the development of a non-continuous moving nested boundary in a
circulation model is novel. Such an innovative solution does not pose restrictions on the location of nested grids
with regards wetting and drying (as demonstrated by the application to Cork Harbour) and, therefore, allows
flexibility of model setup.
Finally, in the context of urban flood modelling, MSN_Flood's ability to simulate horizontal components of
water velocity is a significant advantage over simpler hydraulic models commonly used in flood modelling; the
complexity of urban topography (buildings, vegetation, walls, roads, embankments, ditches etc) necessitates at
least two-dimensional treatment of surface flows (Cook and Merwade, 2009). Spatial and temporal distribution
of velocity fields is also required for assessment of flood risk to people and property associated with a certain
flood flow magnitude. Thus, this feature will greatly benefit flood hazard management.
Although the modelling framework seems to be the main factor controlling accuracy of model predictions, other
factors such as model resolution, datasets and model parameterization also play a crucial role. In relation to
model topography/bathymetry, these aspects are interconnected and need to be considered jointly. Comparing
the 6m and 2 m grid models it can be seen that results are quite sensitive to the spatial resolution of the model.
The resolution acts as a filter on the model terrain so the model error increases with decreasing spatial
resolution, as the definition of topographic features (walls, hedges etc) are progressively lost from the model
bathymetry. There is a dual effect of this. Firstly, as the resolution becomes less granular the topographic
complexity of high density small features become sub-grid phenomena which then become parameterised
through roughness coefficients. Spatially varying roughness needs to be specified for different terrains, this is
determined based on surface classification (such as land type, vegetation or roads) within model sensitivity and
calibration. Secondly, the loss of larger objects such as buildings makes the model inherently ill-conditioned and
their loss cannot be remedied through modification of roughness coefficient alone. Errors are additionally
amplified by a presence of bias in the topographic data resulting from LIDAR related post-processing
difficulties such as representation of surface objects discussed in Mason et al. (2003).

**5 Conclusions**
In this research, high-resolution multi-scale modelling of coastal flooding due to tides, storm surges and rivers
inflows is performed. A state-of-the-art modelling system, MSN_Flood, for simulation of coastal flood




inundation using dynamic downscaling through a cascade of multiple nested grids, was developed to provide a

methodology for accurate assessment of flood inundation. A comprehensive assessment of the modelling system

was carried out for the coastal city of Cork, which is frequently subject to flooding. A November 2009 extreme

flood event driven by both coastal and fluvial mechanisms was selected as a study case. In its application to

Cork City, the flood model comprises of four dynamically nested grids that resolve the hydrodynamics of Cork

Harbour and Cork City at four different scales: 90m, 30m, 6m and 2m. The urban flood model of 2m horizontal

grid resolution is used to simulate flood water inundation of Cork City.

The main findings from this research divided into two thematic groups are summarised here:

1.  Model computational performance:

(a) The nesting model framework allows the model operation at practically any desired horizontal
resolution, including scales commensurate with resolution of LiDAR data making an optimal use of
such datasets. In the current setup, a four-nest cascade telescopes resolution down to the level of
LiDAR resolution which is sufficient to capture small scale flow features.

(b) The model has no limits as to the number of nesting levels and the numerical stability is maintained
down to the finest resolution.

(c) Computational effort is dictated by the number of nesting levels, the horizontal resolution of each
nested grid and the extents of each nested grid. Nevertheless, at the finest resolution the nested model
was found to be almost as accurate as a single grid model of the same resolution but at 96% saving in
computational cost.

(d) Due to its robust flooding and drying routine, the model maintains numerical stability and accuracy in
any part of the model domain affected by these processes.

(e) Internalisation of the nested boundary through a use of ghost cells combined with a tailored adaptive
interpolation technique permits flooding and drying of the nested boundary creating highly dynamic
moving boundaries. Moreover, the flooding and drying mechanism can approach the boundary of the
nested domain from either upstream or downstream. Nesting with a moving boundary allows
embedding of a child grid model within the parent model in areas where the nested boundary may wet
or dry. This unique feature of MSN_Flood provides a high degree of choice regarding the location of
the boundaries to the nested domain and therefore flexibility in model application. This capability gives
MSN_Flood significant advantages over other models.



2.      Model accuracy:
(f)    The modelling system demonstrates a good capability to accurately determine physical processes at

different spatial scales including mesoscale coastal water circulation (90m) and small scale

hydrodynamics of complex urban floodplains (2m).

(g)    The extent of flood inundation into floodplains of Cork City and maximum water levels reached during

flooding were accurately simulated by the urban flood 2 m grid model.

(h)    Fine horizontal resolution is crucial for accurate assessment of inundation. Comparison of 6m and 2m

grid model $RE_T$ in water levels shows a noticeable reduction in model performance at coarser resolution

over the entire domain and the error is generally greater in the dense street network of urbanized zone.

(i)    The urban flood model provides full characteristics of water levels and flow regimes necessary for

assessment of flood risk to people's safety associated with particular flood water levels and associated

flood water velocities.


To conclude, near-unlimited model resolution, geographically unconstrained (due to wetting and drying) nested
model setup, robust wetting and drying routine, computational efficiency and the capability to simulate both
water elevations and velocity fields, make the MSN_Flood a valuable tool for studying coastal flood inundation.
This research demonstrates that the adopted methodology can be successfully used in applications to coastal
flood modelling including complex urban environments. It can provide, at specific instances of time, accurate
spatial distributions of water elevations and flow magnitudes in inundated areas and can, thus, provide critical
information to assess possible extents of flood inundation, periods of inundation, maximum water elevations
reached and flood wave propagation routes and speeds. Ultimately, it can be directly used for evaluation of
flood risks to the area and indirectly, through some functional relationships, for risk assessment of human safety
and property damage. The methodology explored in this research, when applied in a forecasting sense,
constitutes a high resolution flood warning and planning system that can aid local decision makers targeting
high flood risk areas.

**Acknowledgements**
This publication has emanated from research conducted with the financial support of Science
Foundation Ireland (SFI) under Grant Numbers SFI/12/RC/2302 and SFI/14/ADV/RC3021.



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





Tables

Table 1. Configuration of nested models

| Model | Grid size m | Timestep s | Parent model | Parent-to-model grid ratio |
|---|---|---|---|---|
| Parent grid (PG90) | 90 | 18 | -- | 1:1 |
| Single grid (SG30) | 30 | 6 | PG90 | 1:1 |
| Child grid 1 (CG30) | 30 | 6 | PG90 | 3:1 |
| Child grid 2 (CG06) | 6 | 0.6 | CG30 | 5:1 |
| Child grid 3 (CG02) | 2 | 0.2 | CG06 | 3:1 |
| Single grid (SG02) | 2 | 0.2 | CG06 | 1:1 |



Table 2. Summary of error analyses for PG90 and CG30 models within CG30 model area.

| Error Analyses Parameter | SG30 | |
|---|---|---|
| | **PG90** | **CG30** |
| Water Elevation: | | |
| - $RE_D$ [%] | 5.9 | 1.1 |
| - $AE_D$ [x$10^{-2}$ m] | 8.0 | 1.2 |
| - $RE_T > 1\%$ [%] | 94 | 28 |
| Current Velocity: | | |
| - $RE_D$ [%] | 22.4 | 0.5 |
| - $AE_D$ [x$10^{-3}$ m/s] | 2.70 | 0.13 |
| - $RE_T > 5\%$ [%] | 72 | 4 |






Table 3. Error statistics of water elevations simulated by the CG06 model and measured at Tivoli tidal gauge
station. Heights are in meters

| Code | COR | NSD | RMSD | RMSE | RMSdiff |
|------|------|-------|-------|-------|---------|
| CG06_1 | 0.992 | 1.021 | 0.141 | 0.142 | 0.022 |
| CG06_2 | 0.996 | 1.023 | 0.104 | 0.106 | 0.024 |
| CG06_3 | 0.995 | 1.084 | 0.075 | 0.075 | 0.020 |



Table 4. Error statistics of water elevations at four locations simulated by the CG06 and CG02 models. Heights
are in meters

| Code | COR | NSD | RMSD | RMSE | RMSdiff |
|------|------|-------|-------|-------|---------|
| CG02_1 | 0.995 | 1.033 | 0.080 | 0.111 | -0.081 |
| CG02_2 | 0.997 | 1.014 | 0.109 | 0.195 | -0.181 |
| CG02_3 | 0.998 | 1.045 | 0.056 | 0.076 | -0.064 |
| CG02_4 | 0.999 | 0.999 | 0.006 | 0.006 | 0.000 |













Figures

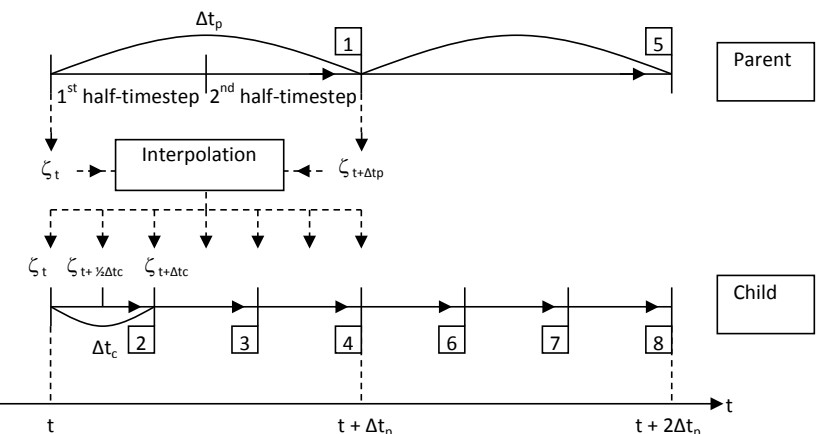

Figure 1: The nesting procedure for a single level of nesting and one variable only - water surface elevation, $\zeta$.


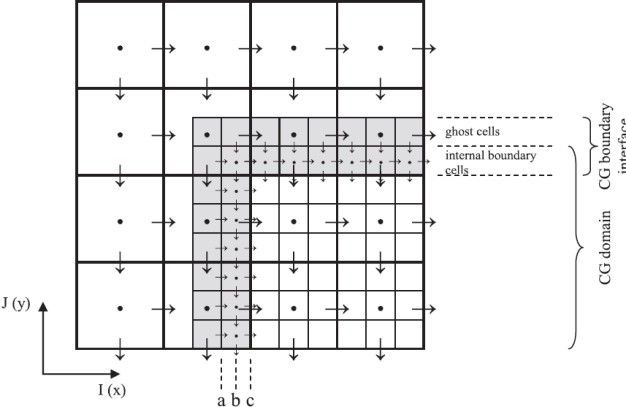


Figure 2: Schematic illustration of the internal boundary configuration for 3:1 nesting ratio.




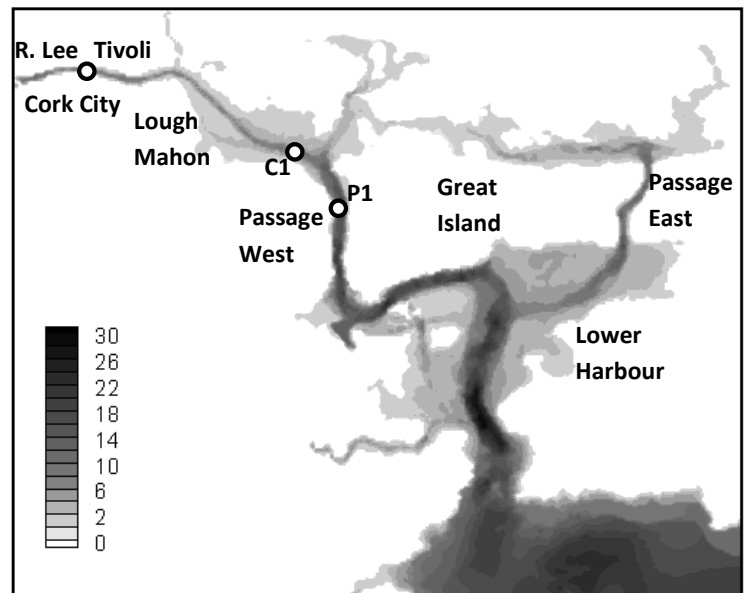

Figure 3: Map of Cork Harbour with selected locations.

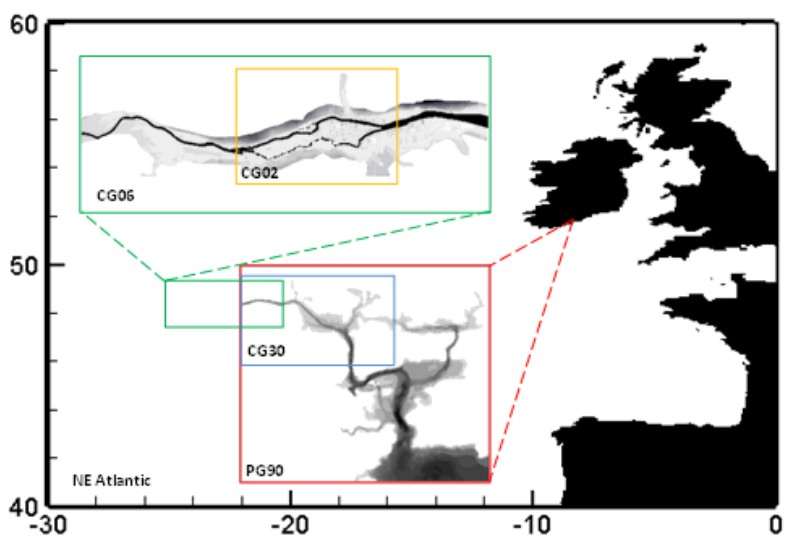


Figure 4: Four-level nesting structure of Cork Harbour and City nested model.





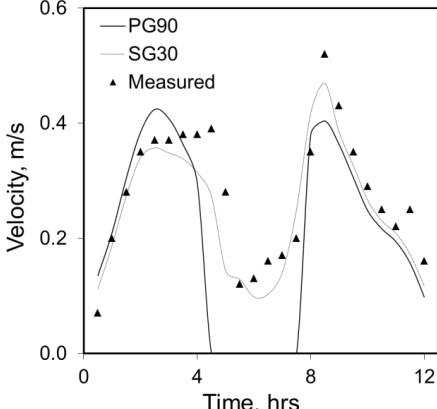


Figure 5: Comparison of computed and measured velocities at Passage West (point C1 in Figure 3).

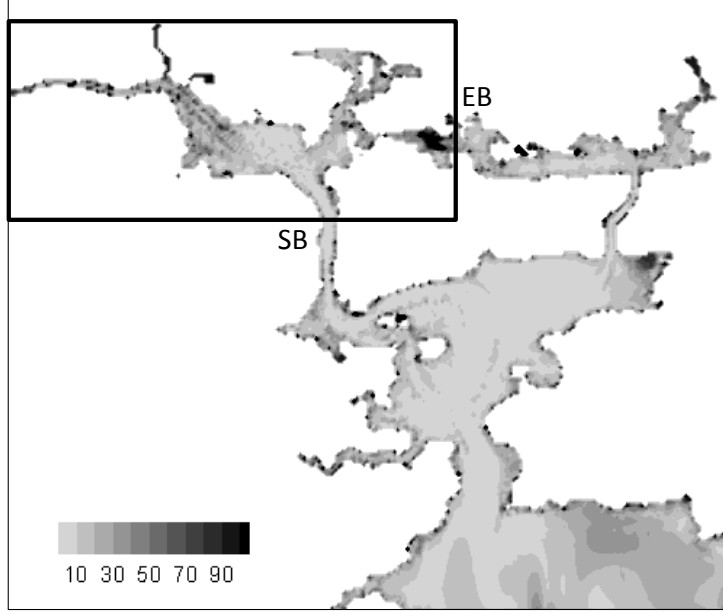

Figure 6: $RE_T$ (%) in  PG90 velocities. Black box shows extents of CG30 model and locations of nested
boundaries. EB - east boundary, SB – south boundary.







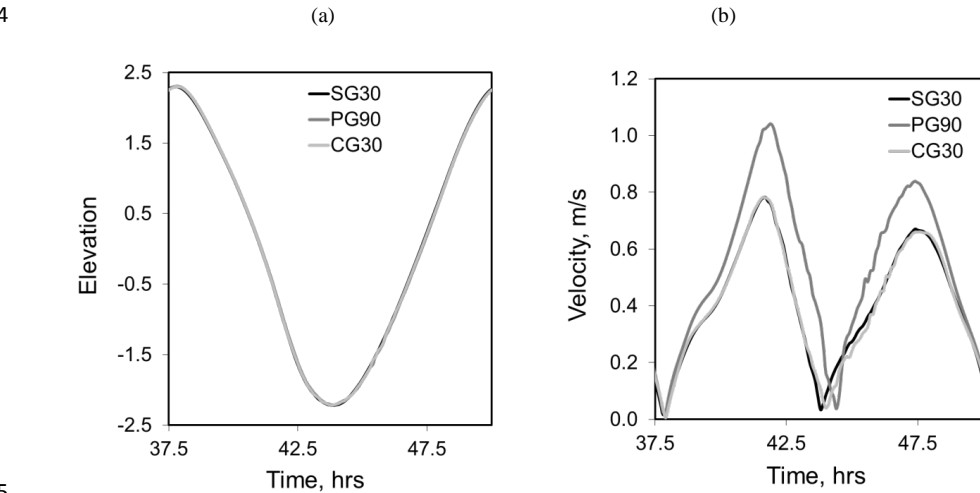

Figure7: Comparison of (a) water elevations and (b) current velocities at point C1 in Lough Mahon.



Figure 8: Comparison of modelled velocities for various grid setups at point C1 in Lough Mahon. Time series
data are overlain by a linear trend.







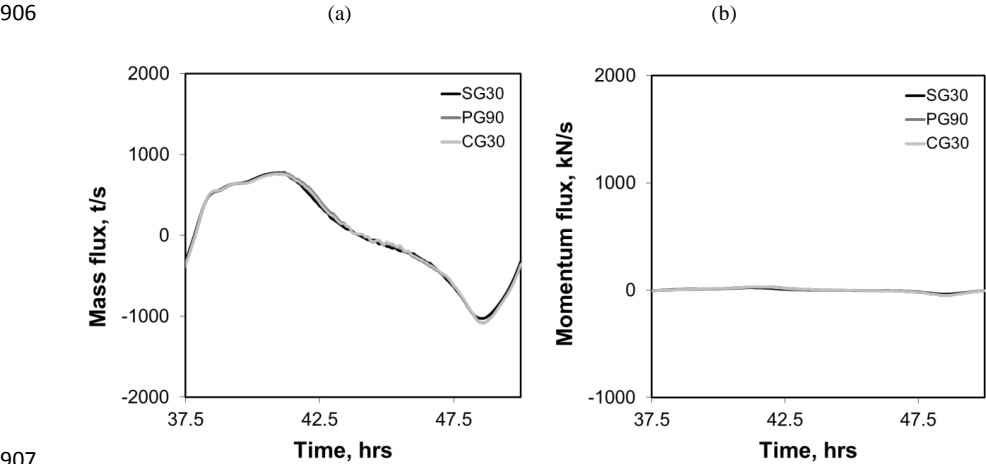


Figure 9: Comparison of (a) mass and (b) momentum fluxes across EB boundary; PG90 and CG30 timeseries
are coincident.


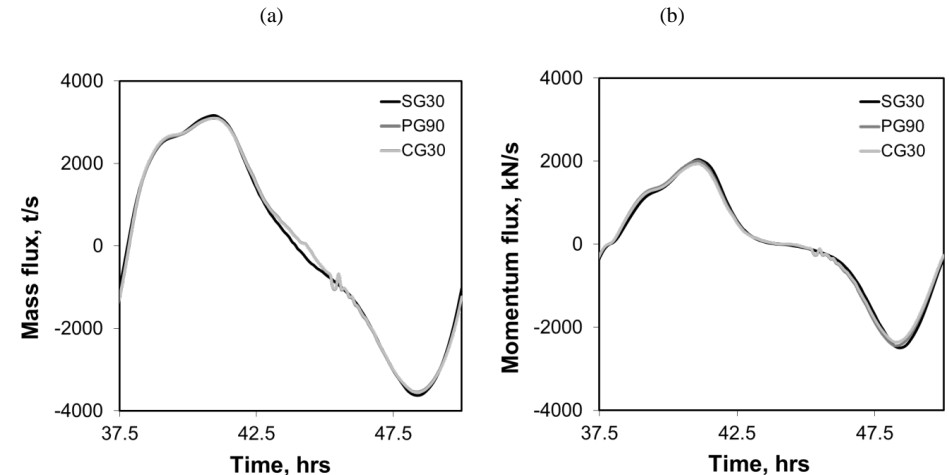


Figure 10: Comparison of (a) mass and (b) momentum fluxes across SB boundary; PG90 and CG30 timeseries
are coincident.




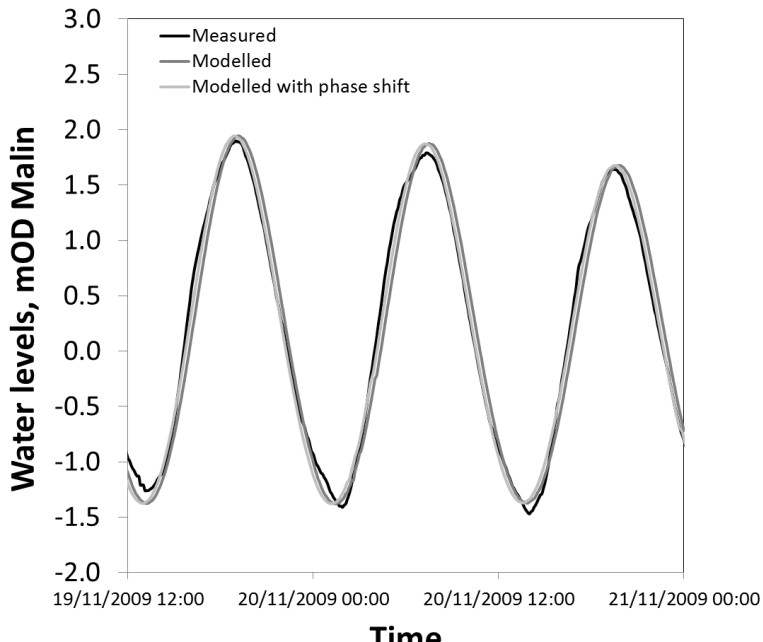


Figure 11: Water elevations predicted by the CG06 model and measured at Tivoli tidal gauge station.


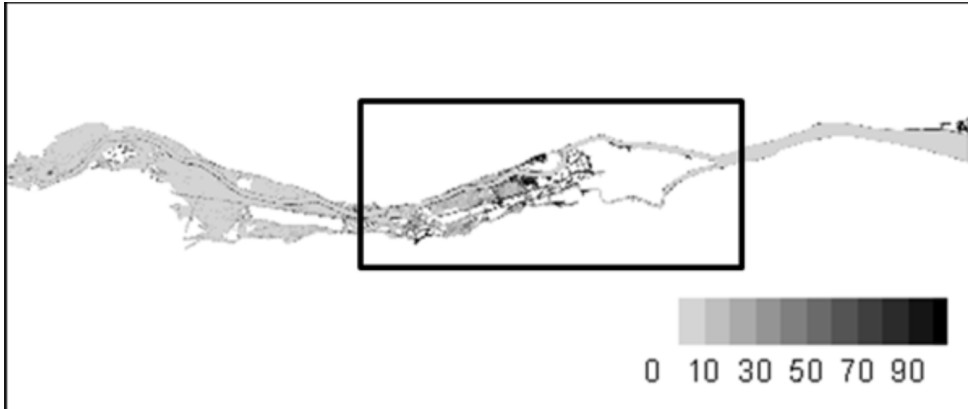


Figure 12: Water level $RE_T$ (%) in CG06 relative to SG02 . Black box shows extents of CG02 model and
locations of nested boundaries.




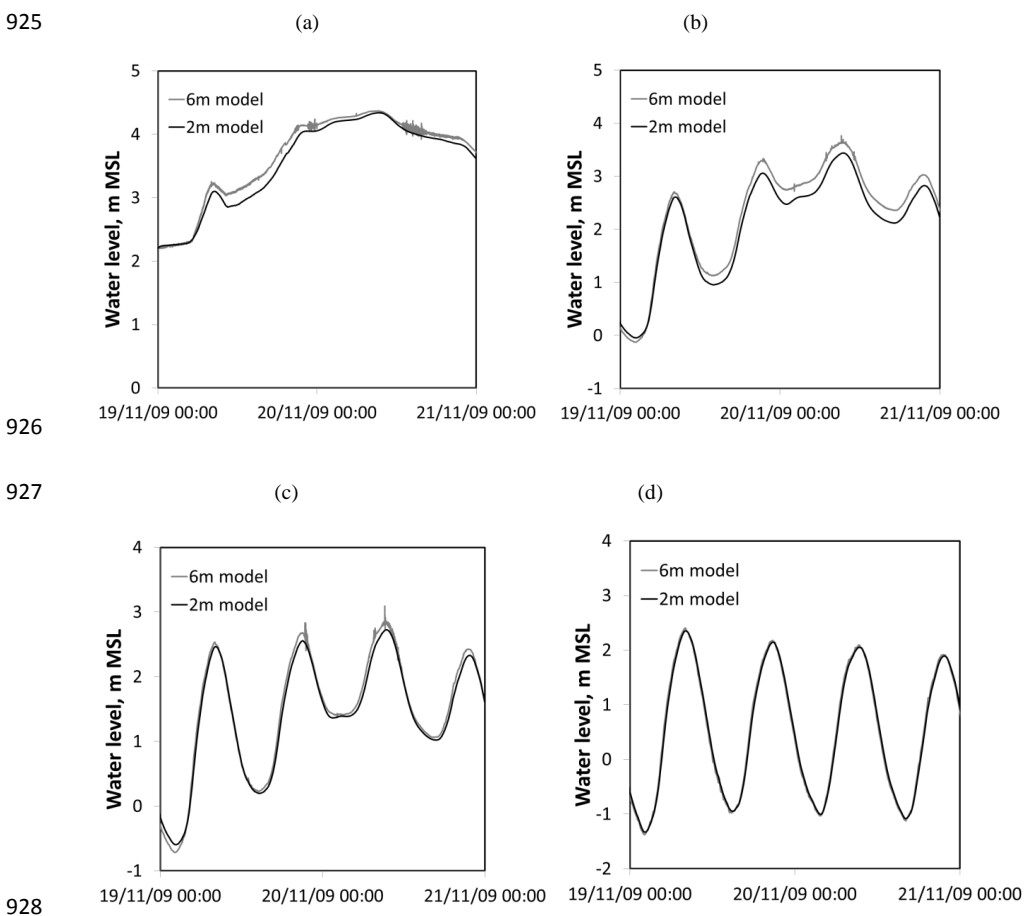

Figure 13: Timeseries of water elevations predicted by CG06 and CG02 models at four locations (a) CG02_1,
(b) CG02_2, (c) CG02_3, (d) CG02_4.




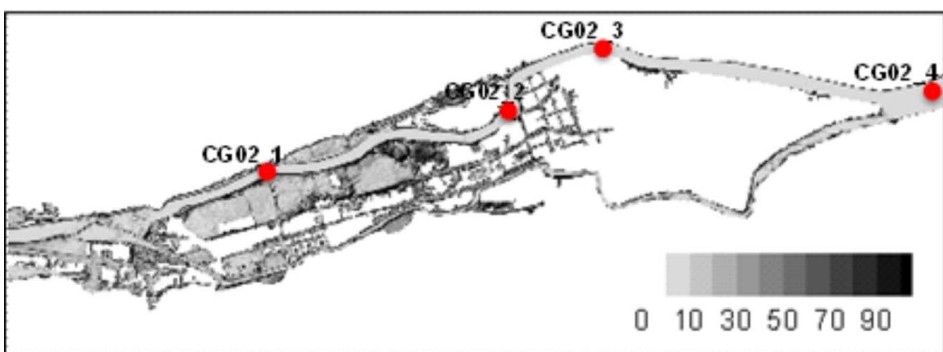


Figure 14: Water level $RE_T$ (%) in CG02 relative to SG02 . Red dots denotes points used in water level analysis

(see Figure 13).

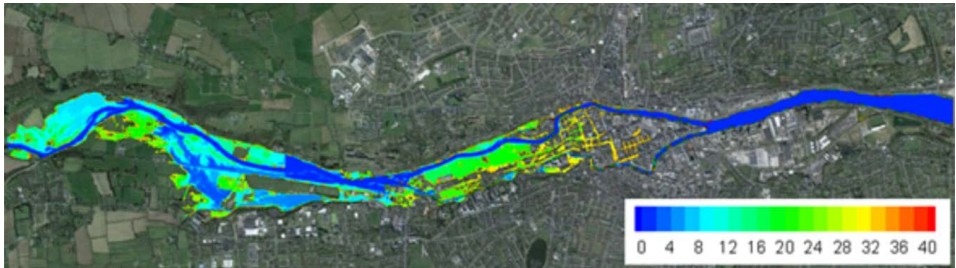


Figure 15: Temporal evolution of flood wave through upper and lower floodplain of Cork City during

November 2009 flood event modelled by CGO6; contours represent 2-hour intervals.




(a)

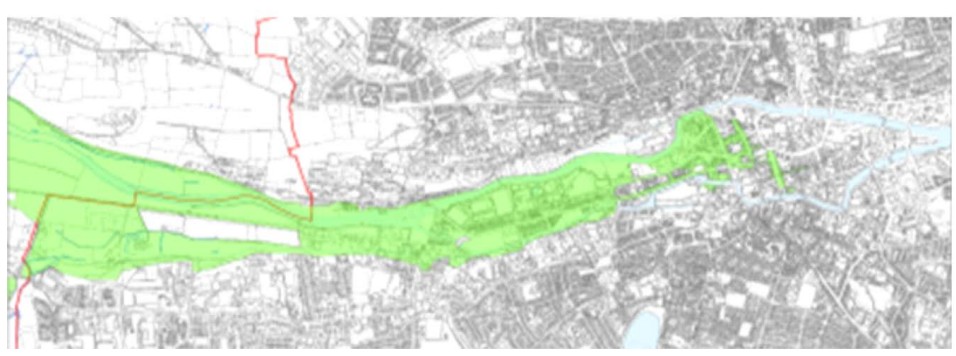

(b)

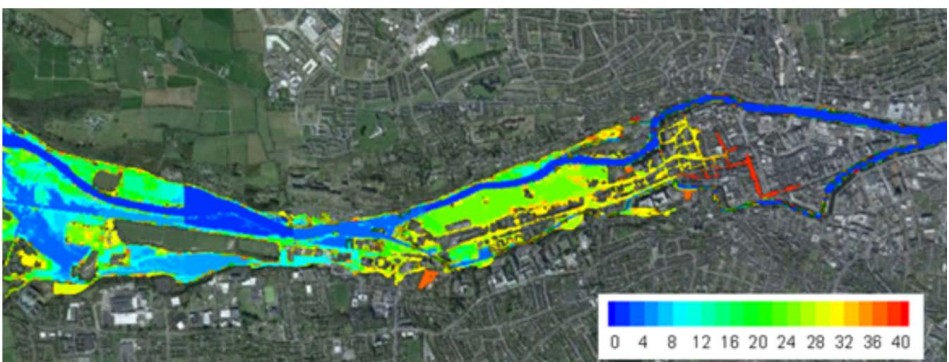


Figure 16: Maps of flood inundation observed by (a) OPW and (b) modelled (contours represent 2-hour
intervals). Evolution of modelled flood wave is a combined output of CG06 and CG02 models.

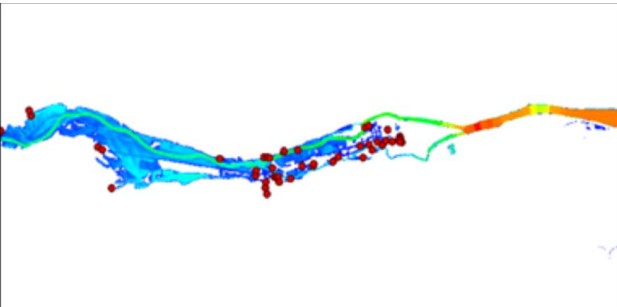


Figure 17: Maximum water levels during November 2009 flood event and water level survey points marked as
red dots.




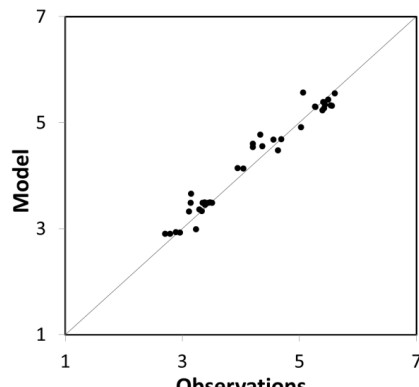


Figure 18: Comparison of modelled and observed maximum water elevations at 38 stations.


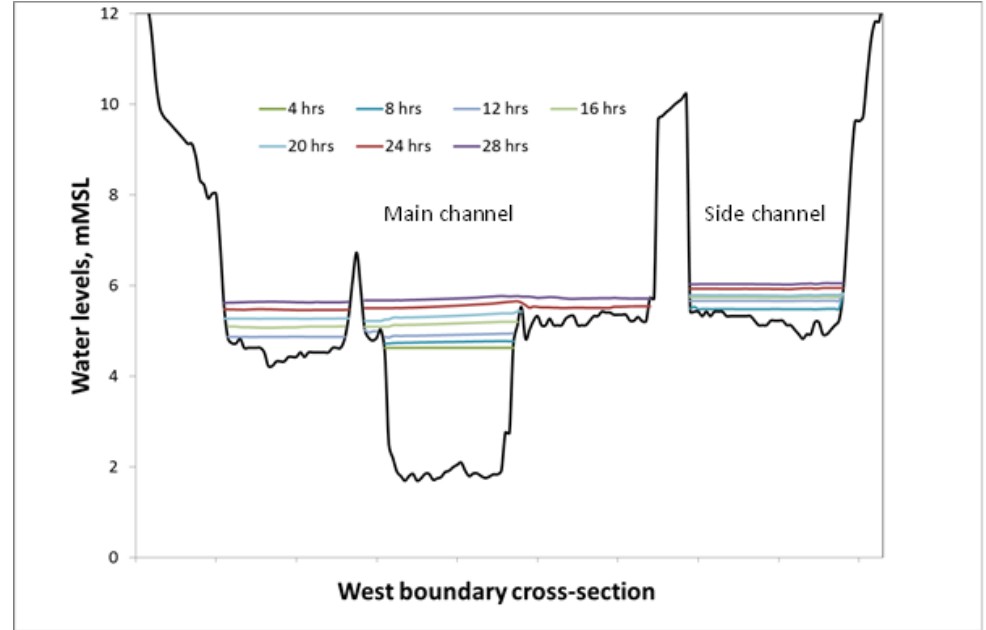


Figure 19: Cross section through west boundary of CG02 model with water elevation marks for selected time

points.





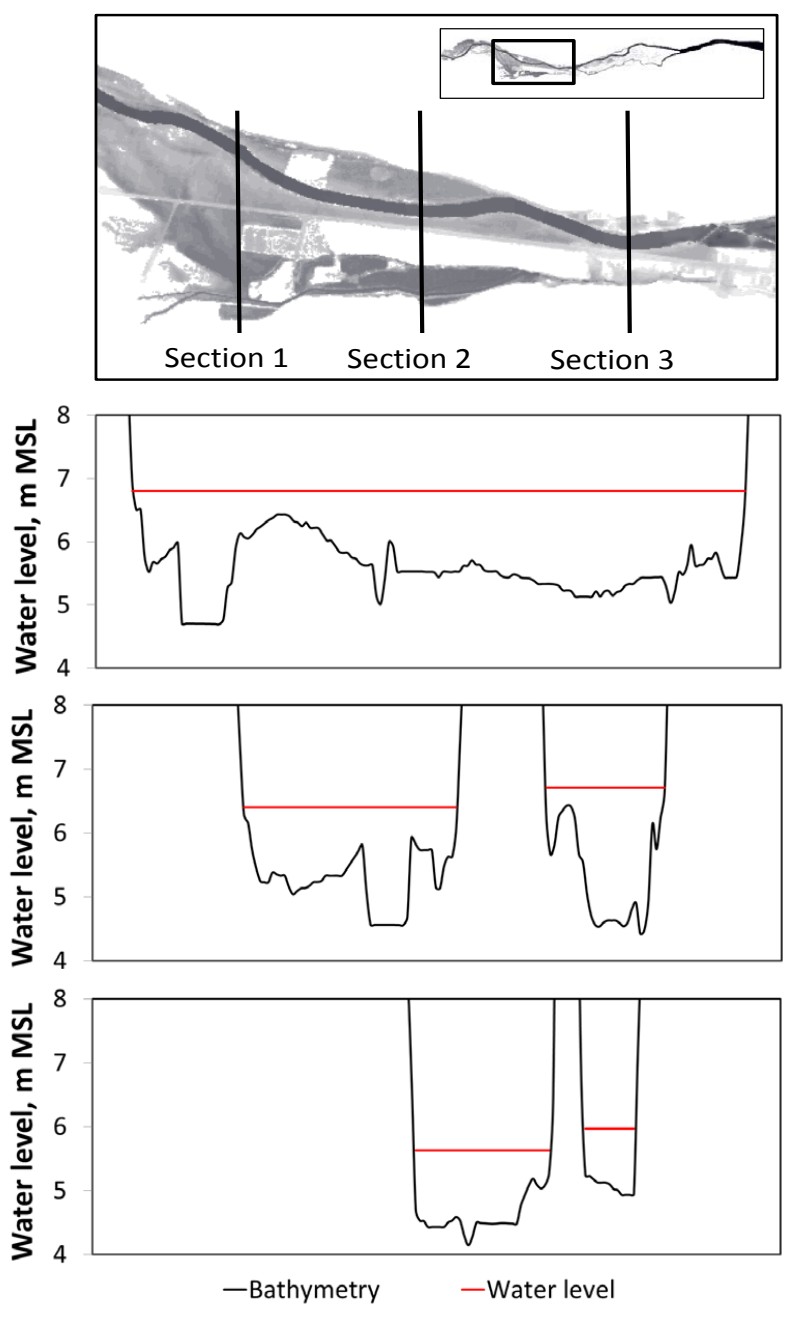


Figure 20: Water elevations at three cross-sections during flooding simulated by CG06.




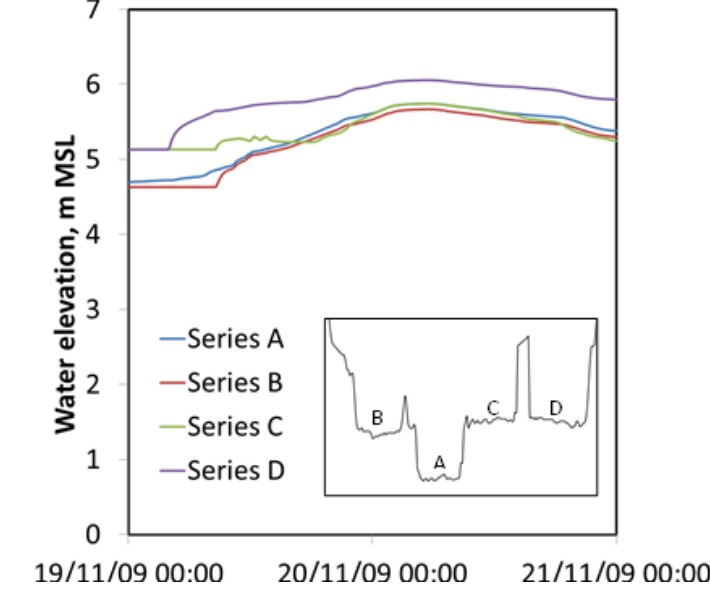



Figure 21: Timeseries of water elevations across the western nested boundary of CG02.




(a)

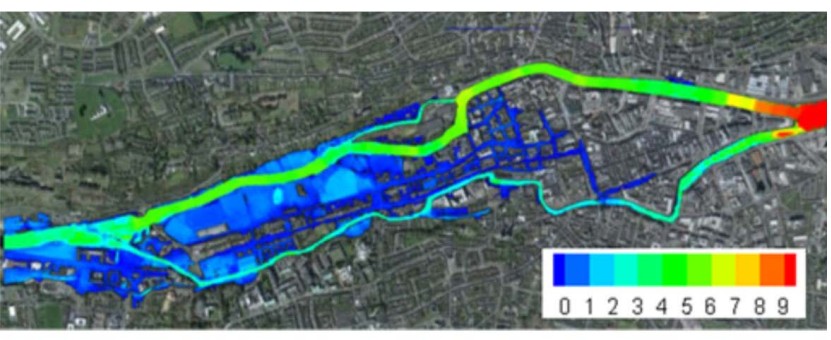

(b)

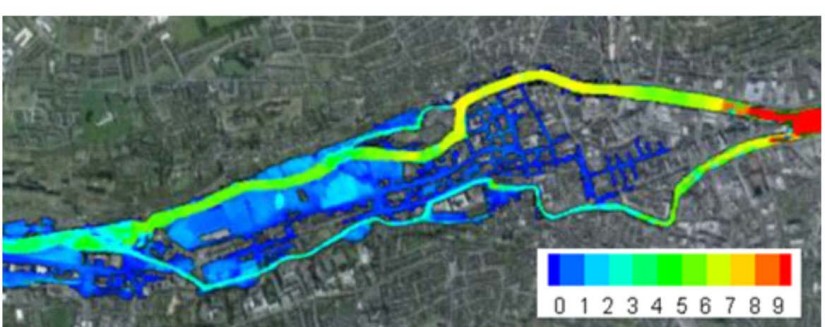


Figure 22:  Comparison of flood extent simulated by  (a) CG02 and  (b) CG06 models. Contours represent water
levels (m).



(a)

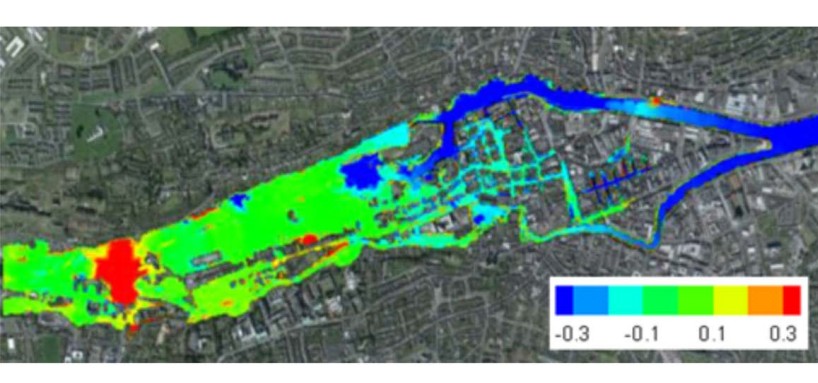

(b)

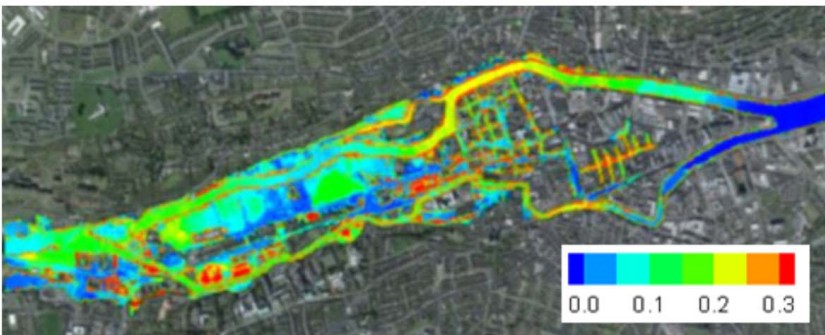


Figure 23: (a) Difference in water elevations (m) between CG06 and CG02 models and (b) RMSE contour plot
over time.












(a)                                                        (b)

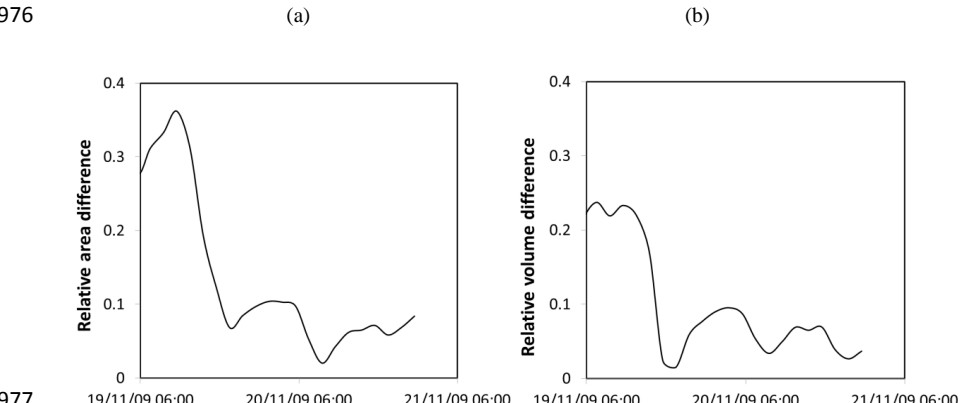


Figure 24: Evolution of the relative difference in  (a) total area of inundation and  (b) volume of water in
inundated area between CG06 and CG02 models. See text for explanation of relative difference.

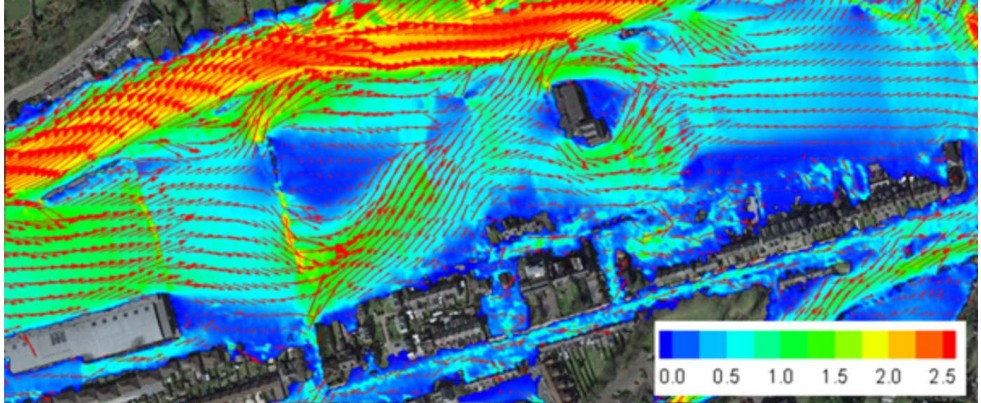


Figure 25: Map of velocity contours (m/s) with vectors showing magnitude and direction of velocities in the
downstream floodplains of Cork City.
