# Peer review of "Development of high-resolution multi-scale modelling system"

_Natural Hazards and Earth System Sciences, 2016_

## Referee Comment (RC1) · Anonymous Referee #1 · 27 Jul 2016

The manuscript proposed multi-scale nested (MSN) Flood to complex coastal-fluvial urban flooding in the estuary-lying Cork City and its capability of the multilevel nesting integrated system to accurately simulate the extent and level of urban flooding was critically examined. The proposed method and model developed in the paper was novel and valid. Therefore this work is suitable for publication after answering some questions. 1. Line 266 to 268: 'The domains of CG30 and CG06 models only partially overlap. Water elevations computed on CG30 are passed to the eastern boundary of CG06 while River Lee flow data are specified at the western boundary of CG06.' So is there any problem of inconsistency between the CG30 data and River Lee flow data? What approach was applied to reduce it?

2. Line 360 to 363 and Figure 7: While the elevation results of PG90 and CG30 are both accurate, further analysis is necessary to explain why accuracy of velocity using PG90 was much lower than that of CG30.

3. Line 423 to 425 and Figure 13: The analysis on infrequent random oscillations in water levels occurring in CG06 (Fig. 13 a-c) should be more detailed.

4. Line 413 to 425 and Table 4: In order to estimate the accuracy of GC02, error statistics of water elevations simulated by the CG02 and measured data should be more suitable than comparing CG06 and CG02 (Table 4). So why use CG06ïij§
* * *

---

## Author Comment (AC1) · 7 Sep 2016

Dear Reviewer,

Thank you very much for reviewing our paper; your valuable comments were much appreciated. We spent a considerable amount of time to address your comments. Please find below our responses:

1. Line 266 to 268: 'The domains of CG30 and CG06 models only partially overlap. Water elevations computed on CG30 are passed to the eastern boundary of CG06 while River Lee flow data are specified at the western boundary of CG06.' So is there any problem of inconsistency between the CG30 data and River Lee flow data? What

approach was applied to reduce it?

CG30 water levels that are specified to CG06 are compared with water levels measured at Tivoli tidal gauge (located on the CG06 eastern boundary) in Figure 11. Very good agreement is achieved showing that at the location of the CG06 boundary, the CG30 model accurately reproduces the water levels recorded during the flood event. Given that the flow data specified at the western boundary of CG06 is for the same time period, the two datasets are therefore consistent.

2. Line 360 to 363 and Figure 7: While the elevation results of PG90 and CG30 are both accurate, further analysis is necessary to explain why accuracy of velocity using PG90 was much lower than that of CG30.

When model grid resolution is too coarse, some flow features such as small-scale gyres may not be accurately resolved or even lost. PG90 does not resolve bathymetry and topography as well as CG30 and therefore produces less accurate flow solutions. This is particularly important in Lough Mahon, which is quite shallow and has complex bathymetry. The errors in Figure 6 are naturally higher in areas of complex bathymetry and coastline where spatial resolution is of most importance.

3. Line 423 to 425 and Figure 13: The analysis on infrequent random oscillations in water levels occurring in CG06 (Fig. 13 a-c) should be more detailed.

The MSN_Flood model used in this research utilises an alternating direction implicit (ADI) algorithm in its solution procedure. The models using ADI are generally very accurate numerically in modelling flows. However, in the presence of a discontinuity, such as a sharp elevation gradient, high elevations or velocity gradients numerical models using such schemes are prone to generate spurious numerical oscillations in the region of sharp gradients (Kvočka et al., 2015). A common solution used to reduce these oscillations is to increase the grid resolution so the slopes over numerical grids are milder. Comparing time series outputs from CG06 and CG02 it is evident that increasing resolution of the model significantly reduces numerical errors and hence

oscillations. This response will be incorporated to the manuscript. Kvočka, D, Falconer, RA, Bray, M (2015) Appropriate model use for predicting elevations and inundation extent for extreme flood events. Natural Hazards 79, 1791-1808

4. Line 413 to 425 and Table 4: In order to estimate the accuracy of GC02, error statistics of water elevations simulated by the CG02 and measured data should be more suitable than comparing CG06 and CG02 (Table 4). So why use CG06ïij§

The only observational data within the extent of CG02 were the maximum water levels during flood event. The comparison of modelled and observed maximum water levels for 38 stations is showed in Figure 18. In the absence of any other measured data CG02 was compared with CG06, which was already showed highly accurate.

---

## Referee Comment (RC2) · Anonymous Referee #2 · 9 Nov 2016

This paper describes a multi-scale nested modelling system to simulate flooding in coastal towns. The study site is Cork City in southern Ireland. The modelling system involves repeated downscaling of coupled numerical models with increasing spatial and temporal resolution, from a relatively coarse coastal ocean model down to very high resolution urban flood model. An innovative feature of the modelling system is the boundary formulation which allows wetting and drying across model boundaries. The authors also convincingly demonstrate that the use of the nested model system provides satisfactory results and is more computationally efficient than running an equivalent high-resolution model for the whole domain. The paper is well written and is suitable for publication after minor revision.

[Figure]

Minor Comments

p. 6, lines 156 – 168: U and V are not specified

p. 7. Line 196: "befits" should be "benefits"

Section 2.3: Presumably the child grids have more refined bathymetry than the parent grids. How are mass and volume conservation achieved when moving from the coarse to the fine grid ?

p. 11, line 311. The definition of errors should be moved to the Methods (Section 2). Similarly for Equations 5 and 6.

p. 14, line 390. What is the RMSD and how is it different from the RMSE ? These errors have not been defined.

p. 15. The "infrequent random oscillations" in CG06 suggest that the model is being run at the limits of numerical stability, presumably to minimise computation time. The authors might improve the results of CG06 by reducing the time step. Does the marginal stability of CG06 affect the quality of the boundary forcing supplied to CG02 ?

p. 17, line 468. "…details of that analysis are presented elsewhere". Where ? Please provide a citation.

p. 17, line 476. This may be a matter of semantics, but I find the use of the term "Moving Boundary" misleading. The boundaries in this model system do not "move" (unless I have missed something), but they are adjustable and variable in extent.

p. 20, line 570 – 571. I think the indices c and f denote CG06 and CG02 respectively, not the other way around.

p. 21, line 585. "oppose" should be "opposed".

p. 23. The Conclusions section is too long and should be shortened. The first paragraph (lines 667 – 675) is a summary, not a conclusion, and could be deleted. The Conclusion should summarise the main findings, starting at line 676.

---

## Author Comment (AC2) · 30 Nov 2016

Response to the Reviewer no 2.

Thank you very much for reviewing our paper and your valuable comments. We appreciate your comments and put our best effort to address them. Please find below our responses:

1. p. 6, lines 156 – 168: U and V are not specified The descriptions of U and V are now placed in the manuscript

2. p. 7. Line 196: "befits" should be "benefits" This is now changed

3. Section 2.3: Presumably the child grids have more refined bathymetry than the parent grids. How are mass and volume conservation achieved when moving from the coarse to the fine grid ? While mass conservation is relatively easily achieved, momentum conservation is more difficult and it is the ghost cell treatment of the nested boundary that ensure we achieve conservation of incoming momentum fluxes. As stated at the top of page 13 (line 348) of the manuscript, the tidally-averaged errors in CG30 fluxes (both mass and momentum) relative to PG90 fluxes were less that 2% at both boundaries, demonstrating high levels of conservation from parent grid to child grid. These boundary flux comparisons are shown in Figure 10.

4. p. 11, line 311. The definition of errors should be moved to the Methods (Section 2). Similarly for Equations 5 and 6. Section 2.5 Verification with statistical definitions and equations has been added to the Methods

5. p. 14, line 390. What is the RMSD and how is it different from the RMSE ? These errors have not been defined. Section 2.5 Verification with statistical definitions and equations has been added to the Methods

6. p. 15. The "infrequent random oscillations" in CG06 suggest that the model is being run at the limits of numerical stability, presumably to minimise computation time. The authors might improve the results of CG06 by reducing the time step. Does the marginal stability of CG06 affect the quality of the boundary forcing supplied to CG02 ? We have analysed the outputs of model CG02 forced with boundary conditions provided by various temporal resolution CG06 model and we found that increasing temporal resolution ofCG06 does not have effect on CG02 model performance but significantly slows down the overall computation time.

7. p. 17, line 468. ". . .details of that analysis are presented elsewhere". Where ? Please provide a citation. Details of the analysis are presented in paper under review which is at the moment in the second round of the review. Since we are not in the position to cite the paper yet, we deleted the statement ". . .details of that analysis are

presented elsewhere" from the manuscript.

8. p. 17, line 476. This may be a matter of semantics, but I find the use of the term "Moving Boundary" misleading. The boundaries in this model system do not "move" (unless I have missed something), but they are adjustable and variable in extent. The "moving boundary" term describes lateral contraction and expansion of the nested boundary. We agree that the term may be misleading but has been widely used to describe this process and we adhered to this terminology. Below are a number of references:

Nash, S., Hartnett, M.: nested circulation modelling of inter-tidal zones: details of nesting approach incorporating moving boundary. Ocean Dynamics 60, 1479-1495, 2010.

Shyy W, Udaykumar HS, Rao MM, Smith RW. Computational Fluid Dynamics with Moving Boundaries. Taylor& Francis: London, 1996. Ahmed, SG, Meshrif, SA (2009) A new numerical algorithm for 2D moving boundary problems using a boundary element method. Computers & Mathematics with Applications 58, 1302-1308 NGA, DDT, Phung, NK (2012) Applying Moving Boundary and Nested Grid to Compute the Accretion, Erosion at the Estuary. Recent Progress in Data Engineering and Internet Technology Volume 157 of the series Lecture Notes in Electrical Engineering pp 1-10

9. p. 20, line 570 – 571. I think the indices c and f denote CG06 and CG02 respectively, not the other way around. Yes, c and f denote CG06 and CG02, respectively. This has been now corrected

10. p. 21, line 585. "oppose" should be "opposed". Yes, this has been now corrected

11. p. 23. The Conclusions section is too long and should be shortened. The first paragraph (lines 667 – 675) is a summary, not a conclusion, and could be deleted. The Conclusion should summarise the main findings, starting at line 676. We shortened the first paragraph of conclusions to two sentences.

---

## Author Response (AR2)

Dear Prof. Tarolli,

We are delighted with your decision on our manuscript acceptance for publication. Thank you very much for editing our paper and your final comments. We have amended figures according to your suggestions. We added units to x-axis in Figures 19 and 20. We also added scale bar to all maps (f03, f06,f12, f14, f15, f16, f17, f20, f22, f23 and f25).
We appreciate all comments from you and anonymous reviewers; they greatly helped to improve the paper.

Once again, thank you very much
Kindest regards

A. Indiana Olbert